# Learning Collaborative Policies to Solve NP-hard Routing Problems

**Minsu Kim**[†]    **Jinkyoo Park**[‡]    **Joungho Kim**[†]
Korea Advanced Institute of Science and Technology (KAIST)
[†] School of Electrical Engineering, [‡] Dept. Industrial & Systems Engineering
`{min-su, jinkyoo.park, joungho}@kaist.ac.kr`

## Abstract

Recently, deep reinforcement learning (DRL) frameworks have shown potential for solving NP-hard routing problems such as the traveling salesman problem (TSP) without problem-specific expert knowledge. Although DRL can be used to solve complex problems, DRL frameworks still struggle to compete with state-of-the-art heuristics showing a substantial performance gap. This paper proposes a novel hierarchical problem-solving strategy, termed learning collaborative policies (LCP), which can effectively find the near-optimum solution using two iterative DRL policies: the seeder and reviser. The seeder generates as diversified candidate solutions as possible (seeds) while being dedicated to exploring over the full combinatorial action space (i.e., sequence of assignment action). To this end, we train the seeder's policy using a simple yet effective entropy regularization reward to encourage the seeder to find diverse solutions. On the other hand, the reviser modifies each candidate solution generated by the seeder; it partitions the full trajectory into sub-tours and simultaneously revises each sub-tour to minimize its traveling distance. Thus, the reviser is trained to improve the candidate solution's quality, focusing on the reduced solution space (which is beneficial for exploitation). Extensive experiments demonstrate that the proposed two-policies collaboration scheme improves over single-policy DRL framework on various NP-hard routing problems, including TSP, prize collecting TSP (PCTSP), and capacitated vehicle routing problem (CVRP).

## 1   Introduction

Routing is a combinatorial optimization problem, one of the prominent fields in discrete mathematics and computational theory. Among routing problems, the traveling salesman problem (TSP) is a canonical example. TSP can be applied to real-world problems in various engineering fields, such as robot routing, biology, and electrical design automation (EDA) [1, 2, 3, 4, 5] by expanding constraints and objectives to real-world settings : coined *TSP variants* are expanded version of TSP. However, TSP and its variants are NP-hard, making it challenging to design an exact solver [6].

Due to NP-hardness, solvers of TSP-like problems rely on mixed-integer linear programming (MILP) solvers [7] and handcrafted heuristics [8, 9]. Although they often provide a remarkable performance on target problems, the conventional approaches have several limitations. Firstly, in the case of MILP solvers, the objective functions and constraints must be formulated into linear forms, but many real-world routing applications, including biology and EDA, have a non-linear objective. Secondly, handcrafted heuristics rely on expert knowledge on target problems, thus hard to solve other problems. That is, whenever the target problem changes, the algorithm must also be re-designed.

Deep reinforcement learning (DRL)-routing frameworks [10, 11, 12] is proposed to tackle the limitation of conventional approaches. One of the benefits of DRL is that reward of DRL can be any

35th Conference on Neural Information Processing Systems (NeurIPS 2021).

value, even from a black-box simulator; therefore, DRL can overcome the limitations of MILP on real-world applications. Moreover, DRL frameworks can automatically design solvers relying less on a handcrafted manner.

We note that the main objective of our research is not outperforming problem-specific solvers like the Concorde [9], a TSP solver. Our problem-solving strategy based on DRL, however, ultimately focuses on practical applications[1] including intelligent transportation [13], biological sequence design [14], routing on electrical device [15] and device placement [16, 17]. Therefore, this paper evaluates the performance of DRL frameworks on TSP-like problems as a benchmark for potential applicability to practical applications, including speed, optimality, scalability, and expand-ability to other problems. TSP-like problems are excellent benchmarks as they have various baselines to compare with and can easily be modeled and evaluated.

**Contribution.** This paper presents a novel DRL scheme, coined learning collaborative policies (LCP), a hierarchical solving protocol with two policies: seeder and reviser. The seeder generates various candidate solutions (seeds), each of which will be iteratively revised by the reviser to generate fine-tuned solutions.

Having diversified candidate solutions is important, as it gives a better chance to find the best solution among them. Thus, the seeder is dedicated to exploring the full combinatorial action space (i.e., sequence of assignment action) so that it can provide as diversified candidate solutions as possible. It is important to explore over the full combinatorial action space because the solution quality highly fluctuates depending on its composition; however, exploring over the combinatorial action space is inherently difficult due to its inevitably many possible solutions. Therefore, this study provides an effective exploration strategy applying an entropy maximization scheme.

The reviser modifies each candidate solution generated by the seeder. The reviser is dedicated to exploiting the policy (i.e., derived knowledge about the problem) to improve the quality of the candidate solution. The reviser partitions the full trajectory into sub-tours and revises each sub-tour to minimize its traveling distance in a parallel manner. This scheme provides two advantages: (a) searching over the restricted solution space can be more effective because the reward signal corresponding to the sub-tour is less variable than that of the full trajectory when using reinforcement learning to derive a policy, and (b) searching over sub-tours of seeds can be parallelized to expedite the revising process.

The most significant advantage of our method is that the reviser can re-evaluate diversified but underrated candidates from the seeder without dropping it out early. Since the seeder explores the full trajectory, there may be a mistake in the local sub-trajectory. Thus, it is essential to correct such mistakes locally to improve the solution quality. The proposed revising scheme parallelizes revising process by decomposing the full solution and locally updating the decomposed solution. Thus it allows the revisers to search over larger solution space in a single inference than conventional local search (i.e., number of iteration of the reviser is smaller than that of conventional local search 2-opt [18], or DRL-based 2-opt [19]), consequently reducing computing costs. Therefore, we can keep the candidates without eliminating them early because of computing costs.

The proposed method is an *architecture-agnostic* method, which can be applied to various neural architectures. The seeder and reviser can be parameterized with any neural architecture; this research utilizes AM [12], the representative DRL model on combinatorial optimization, to parameterize the seeder and the reviser. According to the experimental results, the LCP improves the target neural architecture AM [12], and outperforms competitive DRL frameworks on TSP, PCTSP, and CVRP ($N = 20, 50, 100, 500$, $N$ : number of nodes) and real-world problems in TSPLIB [20]. Moreover, by conducting extensive ablation studies, we show proposed techniques, including entropy regularization scheme and revision scheme, clearly contribute to the performance improvement.

## 2 Related Works

There have been continuous advances in DRL frameworks for solving various routing problems. DRL framework can generate solvers that do not rely on the ground-truth label of target problems: it can be applied to un-explored problems. DRL-based approaches can be categorized into two parts;

---

[1]These works [13, 14, 15, 16, 17] are inspired by DRL frameworks [10, 12] on combinatorial optimization

constructive heuristics and improvement heuristics. We survey these two categories and current emerging hybrid approaches of machine learning (ML) with conventional solvers.

## 2.1 DRL-based Constructive Heuristics

Bello et al. [10] introduced an actor-critic algorithm with a policy parameterized by the pointer network [21]. They proposed a constructive Markov decision process (MDP), where the action is defined as choosing one of the un-served nodes to visit, given a partial solution; the policy is trained to add a node to provide a complete solution sequentially. Later, DRL-based constructive heuristics were developed to design the architecture of neural networks while preserving the constructive MDP [10]. Khalil et al. [11] proposed a DRL framework with a graph embedding structure. Nazari et al. [22], Duedon et al. [23] and Kool et al. [12] redesigned the pointer network [21] using the transformer [24] and trained it with a policy gradient method [25]. The AM by Kool et al. [12] reports substantial results on various NP-hard routing problems, including TSP, PCTSP, CVRP, and orienteering problem (OP) in high-speed computation.

**AM-variants.** After the meaningful success of the AM, many studies are expanded from the AM. Many engineering fields and industries apply AM into their domain. For example, Liao et al. [4] proposed a routing algorithm for the circuit using AM.

Some researches focus on increasing the performances of AM on classic routing problems like TSP by simple techniques. Kwon et al. [26] proposed the POMO, effective reinforcement learning method for AM. They proposed a new RL baseline that can reduce the training variance of AM using the problem-specific property of TSP and CVRP. In addition, they presented an effective post-processing algorithm for TSP and CVRP. However, their proposed method has a limitation in that it is problem-specific because it uses the domain properties of TSP and CVRP (e.g., their method is limited to be applied to PCTSP.).

Xin et al. [27] proposed AM-style DRL-model, MDAM, for NP-hard routing problems. Their method learns multiple AM decoders and derives various solutions through the multiple decoders. The goal of increasing the solution diversity is similar to our research. However, our study is different where it increases the entropy of a single decoder and improves the mistakes of various solutions through a reviser.

## 2.2 DRL-based Improvement Heuristics

Unlike the constructive MDP, DRL-based improvement heuristics are designed to improve the completed solution iteratively. Most researches on DRL-based improvement heuristics are inspired by classical local search algorithms such as 2-opt [18] and the large neighborhood search (LNS) [28].

Chen et al. [29] proposed a DRL-based local search framework, termed NeuRewriter, that shows a promising performance on CVRP and job scheduling problems. Wu et al. [30], and Costa et al. [31] proposed a DRL-based TSP solver by learning the 2-opt. Their method improves the randomly generated solutions, unlike the method of Chen et al. [29] rewrites a solution given by a conventional heuristic solver. Hottung & Tierney [32] proposed a novel search method of VRP that destroys and repairs a solution repeatedly inspired LNS. Their method gives promising performances on CVRP.

Improvement heuristic approaches generally show better performance than constructive heuristics but are usually slower than constructive heuristics. In the case of TSP, the number of neural network's inferences of constructive heuristics is the same as the number of cities to visit. However, the number of inferences of the improvement heuristics is generally much larger.

## 2.3 Hybrid Approaches with Conventional Solvers

There are several studies on hybrid approaches with conventional solvers having promising performance recently. Lu et al. [33] proposed a hybrid method, where the policy is learned to control improvement operators (handcrafted heuristic). Significantly, they outperforms the LKH3, which is widely considered as mountain to climb in machine learning (ML) communities. Joshi et al. [34] combined graph neural network (GNN) model with the beam search algorithm. They trained the GNN with supervised learning for generating a hit map of candidate nodes. Then trained GNN reduces a searching space for improvement heuristics. Similarly, Fu et al. [35] combined supervised

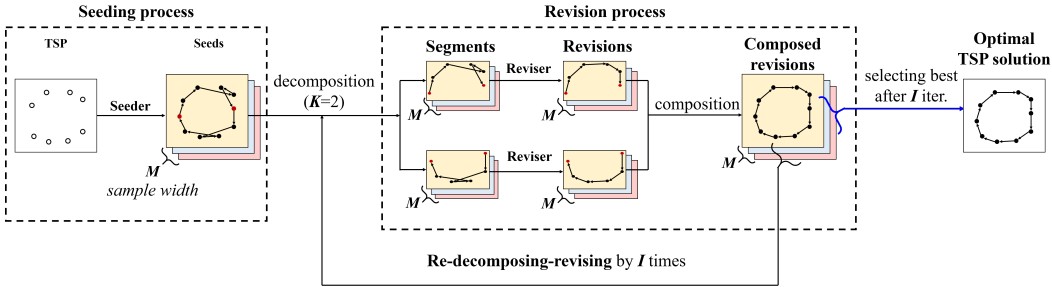

Figure 1: Illustration of seeder-reviser collaboration for TSP.

GNN model with Monte Carlo tree search (MCTS) and Kool et al. [36] combined supervised GNN model with dynamic programming. Their method achieves significant performances, showing ML method can effectively collaborate with conventional operational research (OR) methods.

The research scope of hybrid approaches and DRL-based methods is different. Hybrid approaches can overcome classical solvers in target tasks by collaborating with the classical solvers. However, hybrid approaches have inherited limitations from classical solvers that are poor expandability to other tasks. The DRL-based method can be applied to various real-world tasks without a classic solver; we can also utilize DRL-based to unexplored tasks. This paper investigates the DRL-based NP-hard routing method without the help of classical solvers.

## 3   Formulation of Routing Problems

This section explains the Markov decision process (MDP) formulation for the given 2D Euclidean TSP as a representative example. The formulation of MDP for other problems is described in Appendix A.1.

The main objective of TSP is to find the shortest path of the Hamiltonian cycle. The TSP graph can be represented as a sequence of $N$ nodes in 2D Euclidean space, $\boldsymbol{s} = \{x_i\}_{i=1}^N$, where $x_i \in \mathbb{R}^2$. Then, the solution of TSP can be represented as the permutation $\boldsymbol{\pi}$ of input sequences:

$$\boldsymbol{\pi} = \bigcup_{t=1}^{t=N} \{\pi_t\}, \quad \pi_t \in \{1, ..., N\}, \quad \pi_{t_1} \neq \pi_{t_2} \quad \text{if} \quad t_1 \neq t_2$$

The objective is minimizing the tour length $L(\boldsymbol{\pi}|\boldsymbol{s}) = \sum_{t=1}^{N-1} ||x_{\pi_{t+1}} - x_{\pi_t}||_2 + ||x_{\pi_N} - x_{\pi_1}||_2$.

Then, we formulate the constructive Markov decision process (MDP) of TSP.

**State.** State of MDP is represented as a partial solution of TSP or a sequence of previously selected actions: $\boldsymbol{\pi}_{1:t-1}$.

**Action.** Action is defined as selecting one of un-served tasks. Therefore, action is represented as $\pi_t$ where the $\pi_t \in \{\{1, ..., N\} \setminus \{\boldsymbol{\pi}_{1:t-1}\}\}$.

**Cumulative Reward.** We define cumulative reward for solution (a sequence of assignments) from problem instance $\boldsymbol{s}$ as negative of tourlength: $-L(\boldsymbol{\pi}|\boldsymbol{s})$.

**Constructive Policy.** Finally we define constructive policy $p(\boldsymbol{\pi}|\boldsymbol{s})$ that generates a solution $\boldsymbol{\pi}$ from TSP graph $\boldsymbol{s}$. The constructive policy $p(\boldsymbol{\pi}|\boldsymbol{s})$ is decomposed as:

$$p(\boldsymbol{\pi}|\boldsymbol{s}) = \prod_{t=1}^{t=N} p_\theta(\pi_t|\boldsymbol{\pi}_{1:t-1}, \boldsymbol{s})$$

Where $p_\theta(\pi_t|\boldsymbol{\pi}_{1:t-1}, \boldsymbol{s})$ is a single-step assignment policy parameterized by parameter $\theta$.

## 4   Learning Collaborative Policies

This section describes a novel hierarchical problem-solving strategy, termed learning collaborative policies (LCP), which can effectively find the near-optimum solution using two hierarchical steps,

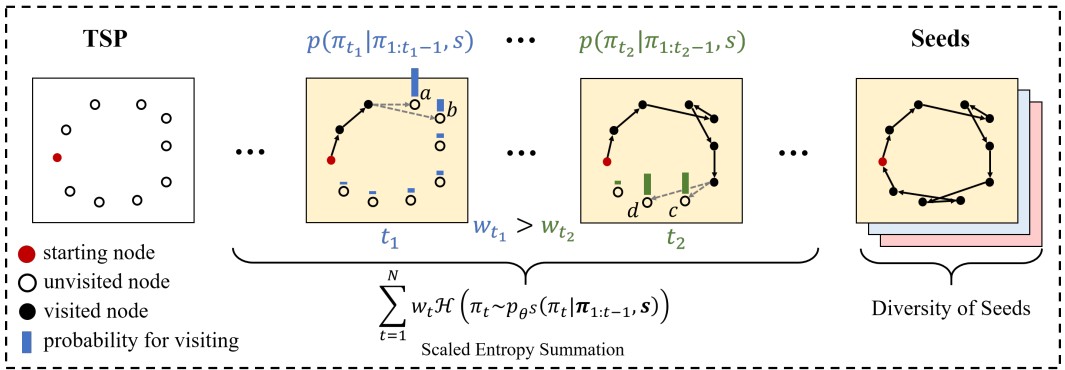

Figure 2: Illustration of the seeding process. The diversity of seeds is approximated as the sum of scaled entropy of segment policy. The scheduler $w_t$ is for giving more weight when $t$ is in early step $t_1$ than $t_2$. In the early step $t_1$, selecting $a$, rather than $b$ is critical for the overall solution. In the latter step $t_2$, even though the probability of selecting node $c$ and $d$ is equal (i.e., has a high entropy), they may give similar solutions.

seeding process and revising process (see Figure 1 for detail). In the seeding process, the seeder policy $p^S$ generates $M$ number of diversified candidate solutions. In the revising process, the reviser policy $p^R$ re-writes each candidate solution $I$ times to minimize the tour length of the candidate. The final solution is then selected as the best solution among $M$ revised (updated) candidate solutions. See pseudo-code in Appendix A.4 for a detailed technical explanation.

## 4.1 Seeding Process

The seeder generates as diversified candidate solutions as possible while being dedicated to exploring the full combinatorial action space. To this end, the seeder is trained to solve the following problems.

**Solution space.** Solution space of seeder is a set of full trajectory solutions : $\{\boldsymbol{\pi}^{(1)}, ..., \boldsymbol{\pi}^{(M)}\}$. The $M$ is the number of candidate solutions from the seeder: termed *sample width*.

**Policy structure.** Seeder is a constructive policy, as defined in section 3 as follows:

$$p^S(\boldsymbol{\pi}|\boldsymbol{s}) = \prod_{t=1}^{t=N} p_{\theta^S}(\pi_t|\boldsymbol{\pi}_{1:t-1}, \boldsymbol{s})$$

The segment policy $p_{\theta^S}(\pi_t|\boldsymbol{\pi}_{1:t-1}, \boldsymbol{s})$, parameterized by $\theta^S$, is derived form AM [12].

**Entropy Reward.** To force the seeder policy $p^S$ to sample diverse solutions, we trained $p^S$ such that the entropy $\mathcal{H}$ of $p^S$ to be maximized. To this end, we use the reward $R^S$ defined as:

$$R^S = \mathcal{H}\left(\boldsymbol{\pi} \sim \prod_{t=1}^{t=N} p_{\theta^S}(\pi_t|\boldsymbol{\pi}_{1:t-1}, \boldsymbol{s})\right) \approx \sum_{t=1}^{N} w_t \mathcal{H}\left(\pi_t \sim p_{\theta^S}(\pi_t|\boldsymbol{\pi}_{1:t-1}, \boldsymbol{s})\right) \qquad (1)$$

The entropy of constructive policy is appropriate for measuring solution diversity. However, computing the entropy of constructive policy is intractable because search space is too large: $N!$. Therefore, we approximate it as a weighted sum of the entropy of segment policies $p_{\theta^S}(\pi_t|\boldsymbol{\pi}_{1:t-1}, \boldsymbol{s})$ evaluated at different time step.

We use a linear scheduler (time-varying weights) $w_t = \frac{N-t}{N_w}$ to boost exploration at the earlier stage of composing a solution; higher randomness imposed by the higher weight $w_t$ at the early stage tends to generate more diversified full trajectories later. The $N_w$ is the normalizing factor, which is a hyperparameter.

**Training scheme.** To train the seeder, we use the REINFORCE [25] algorithm with rollout baseline $b$ introduced by Kool et al. [12]. Then the gradient of each objective function is expressed as follows:

$$\nabla J(\theta^S|\boldsymbol{s}) = E_{\boldsymbol{\pi} \sim p^S}[(L(\boldsymbol{\pi}|\boldsymbol{s}) - \alpha R^S(p_{1:N}^S, \boldsymbol{\pi}) - b(\boldsymbol{s}))\nabla log(p^S)] \qquad (2)$$

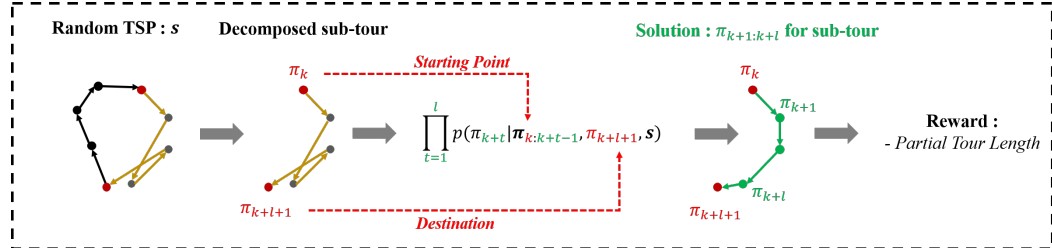

Figure 3: Training progress of the reviser. From randomly generated TSP, sub-problems are decomposed. Then reviser infers to make constructive action referring starting and destination node and previous selected actions. The reward is a negative length of the partial tour generated from the reviser.

Note that the $\alpha$ is hyperparameter for $R^S$ and $p^S_{1:N}$ is the sequence of segment polices $\{p_{\theta^S}(\pi_t|\boldsymbol{\pi}_{1:t-1}, \boldsymbol{s})\}^N_{t=1}$. We use the ADAM [37] optimizer to obtain the optimal parameter $\theta^*$ that minimizes the objective function.

## 4.2   Revision Process

In the revision process, given a candidate solution, the reviser decomposes the candidate solution into $K$ segments and simultaneously finds the optimum routing sequence for each segment of each candidate solution. The reviser repeats this revising process $I$ times to find the best-updated candidate solution. To be specific, the reviser sequentially updates candidate solutions ($I$ times) by repeatably decomposing the full trajectories computed from the previous iteration into segments and revising the segments to produce $M$ updated full trajectory solutions. To sum up, reviser solves $M \times K$ segments in parallel ($M$: number of candidate solutions, $K$: number of the segment in each candidate solution), $I$ times in series.

The proposed scheme has advantages over conventional local search methods or DRL-based improvement heuristics. It searches larger solution spaces in a single inference; therefore, it reduces iteration $I$. For example, 2-opt and DRL-2opt [19] search $O(N^2)$ solution space (if it is parallelizable, $O(MN^2)$), while the reviser searches $O(MK \times l!)$ which is much larger (when the number of nodes of the segment $l$ is big enough) in a single inference. Hence we can reduce the number of iterations $I$ significantly compared to 2-opt, or DRL-2opt [19], thus expediting the speed of the solution search (see Appendix E).

**Solution space.** Solution space of reviser is a partial segment of full trajectory solution represented as $\boldsymbol{\pi}_{k+1:k+l}$. The $k$ is starting index, and $l$ is the number of nodes of the segment. For details of assigning segment including $k$ and $l$, see Appendix A.3.

**Policy structure.** Reviser is a constructive policy as follows:

$$p^R(\boldsymbol{\pi}_{k+1:k+l}|\boldsymbol{s}) = \prod_{t=1}^{t=l} p_{\theta^R}(\pi_{k+t}|\boldsymbol{\pi}_{k:k+t-1}, \pi_{k+l+1}, \boldsymbol{s})$$

The segment policy $p_{\theta^R}$, parameterized by $\theta^R$, is in the similar form with that of AM [12]. Each $\pi_k$ and $\pi_{k+l+1}$ indicate the starting point and the destination point of the partial segment, respectively (see red-points in Figure 3).

We modify the context embedding vector $h^{(N)}_{(c)} = [\bar{h}^{(N)}, h^{(N)}_{\pi_{t-1}}, h^{(N)}_{\pi_1}]$ of AM, which is designed for solving TSP. Hence, $h$ is a high dimensional embedding vector from the transformer-based encoder, and $N$ is the number of multi-head attention layers. $\bar{h}^{(N)}$ is the mean of the entire embedding, $h^{(N)}_{\pi_{t-1}}$ is the embedding of previously selected nodes, and $h^{(N)}_{\pi_1}$ is the embedding of the first node. However, since the destination of reviser is $\pi_{k+l+1}$, not the first node $\pi_1$ , we change the embedding of the first node $h^{(N)}_{\pi_1}$ to be the embedding of the last node $h^{(N)}_{\pi_{k+l+1}}$ for the context embedding as $h^{(N)}_{(c)} = [\bar{h}^{(N)}, h^{(N)}_{\pi_{t-1}}, h^{(N)}_{\pi_{k+l+1}}]$.

**Revision Reward:** negative of partial tour length $L^R(\boldsymbol{\pi}_{k+1:k+l}|\boldsymbol{s}) = \sum_{t=1}^{l+1} ||x_{\pi_{k+t}} - x_{\pi_{k+t-1}}||_2$.

**Training scheme.** The training process is mostly the same as described in section 4.1, except that we have modified the length term $L$ to $L^R$, and set $\alpha = 0$ to remove entropy reward $R^S$ for training the reviser. Note that the seeder and reviser are trained separately.

## 5 Experiments

This section reports the experimental results[2] of the LCP scheme on TSP, PCTSP, and CVRP ($N = 20, 50, 100, 500$, $N$: number of nodes). Also, we report several ablation studies in section 5.3 and Appendix B-F. We evaluate performance on real-world TSPs in the TSPLIB in Appendix G.

**Training Hyperparamters.** Throughout the entire training process of the seeder and reviser, we have exactly the same hyperparameters as Kool et al. [12], except that the training batch size of our seeder is 1024. To train the seeder's policy, we set $\alpha = 0.5$ (2) and $N_w = \sum_{i=1}^{N} i$ for linear weight $w_t = \frac{N-t}{N_w}$ for entropy scheduling.

Details in the experimental setting, including hyperparameters, dataset configuration, and run time evaluation, are described in Appendix A.5.

### 5.1 Target Problems and Baselines

We evaluate the performance of LCP in solving the three routing problems: TSP, PCTSP, and CVRP. We provide a brief explanation of them. The detailed descriptions for these problems are in Appendix A.1.

**Travelling salesman problem (TSP).** TSP is a problem to find the shortest Hamiltonian cycle given node sequences.

**Price collecting travelling salesman problem (PCTSP).** PCTSP [38] is a problem, where each node has a prize and a penalty. The goal is to collect the nodes with at least a minimum total prize (constraint) and minimize tour length added with unvisited nodes' penalties.

**Capacitated vehicle routing problem (CVRP).** CVRP [39] is a problem where each node has a demand, while a vehicle must terminate the tour when the total demand limit is exceeded (constraint). The objective is to minimize the tour length.

For the baseline algorithms, we use two types of algorithms: conventional heuristics and DRL-based solvers. For the conventional heuristics, we use Gurobi [7] (the commercial optimization solver), and the OR Tools [40] (the commercial optimization solver) for all three problems. In Table 1, Gurobi ($t$) indicates time-limited Gurobi whose running time is restricted below $t$. In addition, OR Tools ($t$) is the OR Tools that allows additional local search over a duration of $t$. For problem-specific heuristics, we use Concorde [9] for TSP, the iterative local search (ILS) [12] for PCTSP, and LKH3 [41] for CVRP.

For the baselines using DRL-based solvers, we concentrated on the ability of the LCP scheme, which is improved performance over AM. Validating that the two-policies collaboration scheme outperforms the single-policy scheme (i.e., AM) is a crucial part of this research; thus, the most important metric for performance evaluation is improvement between vanilla AM the AM + LCP. Also, we reproduced other competitive DRL frameworks: current emerging improvement heuristics. We exclude recently proposed AM-style constructive heuristics, including the POMO [26] and MDAM [27] because they can be candidate collaborators with LCP, not competitors (e.g., POMO + LCP is possible). The detailed method for evaluation baselines in Table 1 is described as follows:

**TSP.** We follow baseline setting of Kool et al. [12] and Costa et al. [19]. We set DRL baselines including the S2V-DQN [11], EAN [23], GAT-T [30], DRL-2opt [19], and AM [12]. We show the results of S2V-DQN and EAN reported by Kool et al. [12], and the results of GAT-T reported by Costa et al. [19]. Then we directly reproduce the two most competitive DRL frameworks among baselines, the AM and DRL-2opt, in our machine to make a fair comparison of the speed.

**PCTSP.** We follow baseline setting of Kool et al. [12]. We reproduce AM [12] for DRL baseline.

---

[2]See source code in https://github.com/alstn12088/LCP

Table 1: Performance evaluation results of the LCP scheme compared with baseline heuristics and the DRL frameworks on TSP, PCTSP, and CVRP. The best costs (objective) among DRL frameworks are marked in bold. The *H* is heuristic and *Solver* is exact algorithm. The "LCP {640,2}" means that the sampling width $M$ is 640, and the number of iterations $I$ (of the reviser) is 2. We measure the performance in a limited time budget, which is 10 seconds per instance. The *OB* means "out of budget" and *IF* means "infeasible", where the solver cannot generate solutions satisfying constraints of target problems in a limited time.

| Method | | $N = 20$ | | | $N = 50$ | | | $N = 100$ | | |
|---|---|---|---|---|---|---|---|---|---|---|
| | | Cost | Gap | Time | Cost | Gap | Time | Cost | Gap | Time |
| TSP | Gurobi (*Solver*) | 3.84 | 0.00% | 0.01s | 5.70 | 0.00% | 0.01s | 7.76 | 0.00% | 0.02s |
| | OR Tools (*H*) | 3.85 | 0.37% | | 5.80 | 1.76% | | 8.12 | 4.53% | |
| | Concorde (*H*) | 3.84 | 0.00% | 0.01s | 5.70 | 0.00% | 0.01s | 7.76 | 0.00% | 0.02s |
| | S2V-DQN | 3.89 | 1.42% | | 5.99 | 5.16% | | 8.31 | 7.03% | |
| | EAN {$M$: 1280} | 3.84 | 0.11% | | 5.77 | 1.28% | | 8.75 | 12.7% | |
| | EAN+2OPT {$M$: 1280} | 3.84 | 0.09% | | 5.75 | 1.00% | | 8.12 | 4.64% | |
| | GAT-T {$I$: 5000} | 3.84 | 0.00% | | 5.71 | 0.20% | | 7.87 | 1.42% | |
| | Drl-2opt {$I$: 2000} | 3.84 | 0.00% | 3.58s | 5.70 | 0.12% | 4.88s | 7.83 | 0.87% | 7.15s |
| | AM {$M$: 1280} | 3.84 | 0.08% | 0.03s | 5.73 | 0.52% | 0.09s | 7.94 | 2.26% | 0.36s |
| | AM {$M$: 2560} | 3.84 | 0.06% | 0.04s | 5.72 | 0.45% | 0.13s | 7.94 | 2.21% | 0.42s |
| | AM {$M$: 7500} | 3.84 | 0.05% | 0.06s | 5.72 | 0.39% | 0.29s | 7.93 | 2.13% | 1.21s |
| | AM + **LCP** {640,10} | **3.84** | 0.00% | 0.18s | 5.70 | 0.13% | 0.30s | 7.86 | 1.25% | 0.57s |
| | AM + **LCP** {1280,10} | - | | | 5.70 | 0.10% | 0.45s | 7.85 | 1.13% | 0.90s |
| | AM + **LCP*** {1280,45} | - | | | **5.70** | 0.02% | 2.48s | **7.81** | 0.54% | 4.30s |
| PCTSP | Gurobi (*Solver*) | 3.13 | 0.00% | 0.01s | | *OB* | | | *OB* | |
| | Gurobi {1s} (*H*) | 3.14 | 0.07% | 0.01s | | *IF* | | | *IF* | |
| | Gurobi {10s} (*H*) | 3.13 | 0.00% | 0.01s | 5.17 | 15.6% | 0.19s | | *IF* | |
| | OR Tools {10s} (*H*) | 3.14 | 0.05% | 0.31s | 4.51 | 0.70% | 0.31s | 6.35 | 6.21% | 0.31s |
| | OR Tools {60s} (*H*) | 3.13 | 0.01% | 1.80s | 4.48 | 0.00% | 1.80s | 6.08 | 1.56% | 1.80s |
| | ILS (*H*) | 3.16 | 0.77% | 0.10s | 4.50 | 0.67% | 0.72s | 5.98 | 0.00% | 4.32s |
| | AM {$M$: 1280} | 3.18 | 0.39% | 0.03s | 4.52 | 0.74% | 0.07s | 6.08 | 1.67% | 0.17s |
| | AM {$M$: 2560} | 3.15 | 0.41% | 0.03s | 4.51 | 0.72% | 0.10s | 6.07 | 1.57% | 0.28s |
| | AM + **LCP** {640,1} | 3.14 | 0.17% | 0.04s | 4.50 | 0.51% | 0.07s | 6.06 | 1.42% | 0.15s |
| | AM + **LCP** {1280,5} | **3.14** | 0.08% | 0.10s | **4.49** | 0.32% | 0.20s | **6.04** | 1.00% | 0.39s |
| CVRP | Gurobi (*Solver*) | 6.10 | 0.00% | 0.01s | | *OB* | | | *OB* | |
| | OR Tools (*H*) | 6.43 | 5.41% | | 11.31 | 9.01% | | 17.16 | 9.67% | |
| | LKH3 (*H*) | 6.14 | 0.58% | 0.72s | 10.38 | 0.00% | 2.52s | 15.65 | 0.00% | 4.68s |
| | RL {$M$: 10} | 6.40 | 4.92% | | 11.15 | 7.46% | | 16.96 | 8.39% | |
| | NLNS {$I$: 2000} | 6.19 | 1.47% | 1.00s | 10.54 | 1.54% | 1.63s | 16.00 | 2.17% | 2.18s |
| | AM {$M$: 1280} | 6.25 | 2.49% | 0.05s | 10.62 | 2.40% | 0.14s | 16.23 | 3.72% | 0.34s |
| | AM {$M$: 2560} | 6.25 | 2.39% | 0.06s | 10.61 | 2.24% | 0.31s | 16.17 | 3.34% | 0.75s |
| | AM {$M$: 7500} | 6.24 | 2.24% | 0.09s | 10.59 | 2.06% | 0.36s | 16.14 | 3.11% | 1.42s |
| | AM + **LCP** {640,1} | 6.17 | 1.15% | 0.07s | 10.56 | 1.74% | 0.15s | 16.05 | 2.58% | 0.30s |
| | AM + **LCP** {1280,1} | 6.16 | 0.92% | 0.09s | 10.54 | 1.54% | 0.20s | 16.03 | 2.43% | 0.45s |
| | AM + **LCP** {2560,1} | **6.15** | 0.84% | 0.14s | **10.52** | 1.38% | 0.31s | 16.00 | 2.24% | 0.77s |
| | AM + **LCP** {6500,1} | - | | | - | | | **15.98** | 2.11% | 1.73s |

**CVRP.** We follow baseline setting of Houttung & Tierney [32]. We report result of RL [22] based on Houttung & Tierney [32] and we reproduce AM [12] and NLNS [32].

## 5.2 Performance Evaluation

In this section, we report the performance of LCP on small-scale problems ($N = 20, 50, 100$) in Table 1. Then we provide a time-performance trade-off analysis including large-scale problems ($N = 500$). We note that time-performance analysis is significant because any method can find an optimal solution when given an infinite time budget. From the analysis, we can identify a specific time region, called *winner region*, where LCP performs the best in terms of both speed and performance.

**Performance evaluation on $N = 20, 50, 100$.** Our method outperforms all the DRL baselines and OR-tools in TSP, PCTSP, and CVRP, as clearly shown in Table 1.

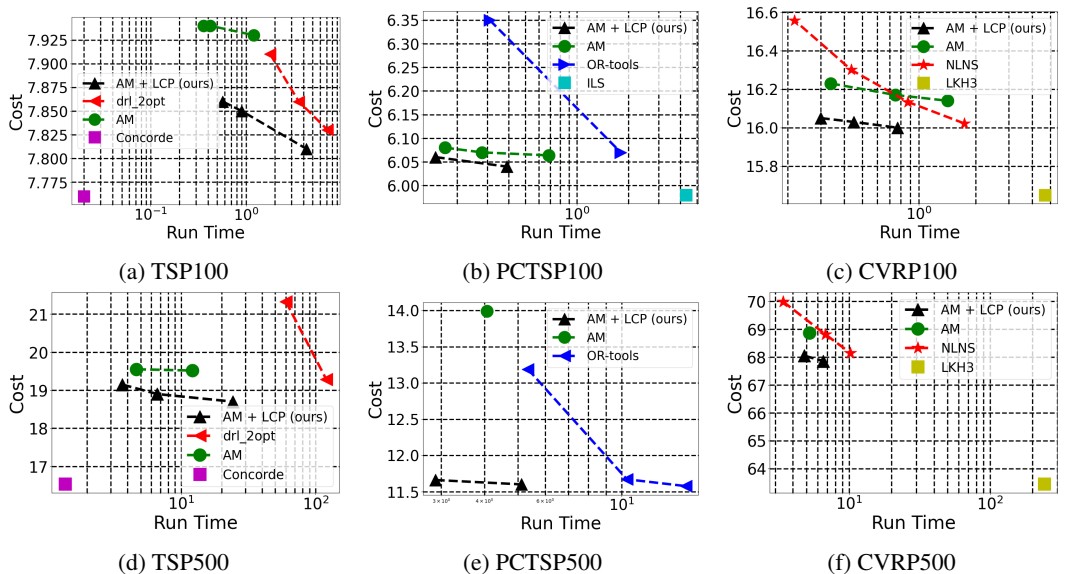

Figure 4: Details of time-performance analysis. The left (fast) and lower (lower cost) trend indicates a better performance. In (e), the ILS cannot generate solution in time budget ($t < 1000s$)

Table 2: Ablation study of LCP components on TSP, PCTSP, and CVRP ($N = 100$). The optimal gap is measured by comparing it with state-of-the-art solvers. The Entropy Regularization indicates training the seeder with $R^S$, while the default is the uniform scheduling. The best performances are marked in bold.

| Component of the LCP | | | TSP | | PCTSP | | CVRP | |
|---|---|---|---|---|---|---|---|---|
| Entropy Regularization | Weight Scheduling | Reviser | cost | gap | cost | gap | cost | gap |
| | | | 7.96 | 2.65% | 6.08 | 1.64% | 16.29 | 3.43% |
| ✓ | | | 7.96 | 2.68% | 6.08 | 1.76% | 16.25 | 3.16% |
| ✓ | ✓ | | 7.94 | 2.45% | 6.07 | 1.62% | 16.20 | 2.86% |
| | | ✓ | 7.86 | 1.32% | 6.04 | 1.13% | 16.20 | 2.86% |
| ✓ | | ✓ | 7.84 | 1.17% | 6.05 | 1.16% | 16.16 | 2.59% |
| ✓ | ✓ | ✓ | **7.82** | **0.88%** | **6.04** | **1.02%** | **16.12** | **2.37%** |

Note that for TSP ($N = 100$), we applied two types of revisers, each of which is denoted LCP and LCP*, respectively. The details are described in Appendix A.4 with pseudo-code. Our LCP and LCP* outperforms DRL-2opt, the current state-of-the-art DRL-based improvement heuristic in $N = 20, 50, 100$, surpass 0.33% in $N = 100$.

In PCTSP, LCP outperforms AM with less time. Our method (AM + LCP {640,1}) outperforms the OR-Tools (10s), with $4\times$ and $2\times$ faster speed in $N = 50, 100$ respectively. Compared to the ILS, our method (AM + LCP {1280,5}) underperforms by $1.0\%$, but has $11 \times$ faster speed for $N = 100$ .

For CVRP, our method outperforms competitive DRL frameworks.

**Time-performance analysis on $N = 100, 500$.** In Figure 4, we describe the time-performance analysis. We cannot control the speed of the Concorde, ILS, and LKH3. We can control the speed of DRL solvers by adjusting sample width $M$ or the number of iterations $I$. For PCTSP, we can change the speed of OR-tools by managing the time for additional local searches.

Our scheme clearly outperforms DRL-solvers in terms of both speed and performance. For PCTSP ($N = 100, 500$) and CVRP ($N = 500$), our method achieves the *winner region* of $t < 10$, which is best performed in a specific time region among all kind of baseline solvers (for CVRP ($N = 100$), our method achieves the *winner region* of $t < 5$).

**Performance on TSPLIB [20] data:** see Appendix G.

### 5.3  Ablation Study

In this section, we conduct an ablation study on LCP components. We leave further ablation studies to Appendix B-F.

**Ablation study of collaborative policies.** In Table 2, we ablate three significant components of LCP and show the experimental results for every case. In the case of vanilla AM, having none of LCP components, the performance is the poorest. On the other hand, collaboration of seeder trained with linearly scheduled-entropy and the reviser shows the best performance. Therefore, the experimental results empirically validate our proposal of hierarchically collaborating two policies and also demonstrate the effectiveness of using a linearly scheduled-entropy term shown in section 4.1 and Figure 2.

**Ablation study of entropy regularization:** see Appendix B.

**Ablation study of SoftMax temperature:** see Appendix C.

**Ablation study of application of LCP to pointer network [10, 21]:** see Appendix D.

**Comparison with reviser and other improvement heuristics:** see Appendix E.

**Training convergence of seeder and reviser in different PyTorch seeds:** see Appendix F.

## 6  Discussion

In this paper, we proposed a novel DRL scheme, learning collaborative policies (LCP). The extensive experiments demonstrate that our two-policies collaboration algorithm (i.e., LCP) outperforms conventional single-policy DRL frameworks, including AM [12], on various NP-hard routing problems, such as TSP, PCTSP, and CVRP.

We highlight that LCP is a reusable scheme, can solve various problems. The neural architecture of the seeder and reviser proposed in this paper is derived from AM [12]. It can be substituted by other architectures, such as the pointer network [10, 21] and AM-style architectures including POMO [26] and MDAM [27]. If further studies on neural architecture for combinatorial optimization are carried out, the seeder and reviser can be improved further.

Also, LCP can be directly applied to other combinatorial optimization tasks, including TSP with time windows (TSPTW), orienteering problem (OP), multiple TSP (mTSP), variations of the vehicle routing problem (VRP), and other practical applications.

**Further Works.** We made an important first step: two-policies collaboration where each policy specializing in exploration or exploitation can improve conventional single-policy systems on combinatorial optimization tasks. The important direction of further research is introducing more sophisticated strategies to explore or exploit combinatorial solution space. New exploration strategies for overcoming the proposed approximated entropy maximization scheme are needed. Also, it is necessary to investigate more effective exploitation strategies beyond the proposed revision scheme.

## Acknowledgements and Disclosure of Funding

This research is supported in part by the KAIST undergraduates research program (URP), 2019. We thank Hankook Lee and Prof. Jinwoo Shin for building part of this project in the URP. We thank Joonsang Park, Keeyoung Son, Hyunwook Park, Haeyeon Rachel Kim, and our anonymous reviewers for feedback and discussions.

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
