# A  Details of Experiments

## A.1  Detailed Explanation of Target Problems.

This paper solves three NP-hard routing problems, traveling salesman problem (TSP), prize collecting TSP (PCTSP), and capacitated vehicle routing problem (CVRP). This section provides detailed descriptions of PCTSP and CVRP (for TSP, see section 3).

**Prize Collecting Travelling Salesman Problem (PCTSP).** The PCTSP is similar to TSP, while there are differences in that we do not have to visit all the nodes and that the destination is not the first node but the depot node, i.e., a tour is not a cycle. Let $N$ be the number of nodes. The problem instance of PCTSP is $s = \{(x_i, \lambda_i, \mu_i)\}_{i=1}^{N+1}$, where the $x_i \in \mathbb{R}^2$ is in 2D euclidean coordinates, $\lambda_i \in \mathbb{R}$ is the penalty of unvisited node, and $\mu_i \in \mathbb{R}$ is the prize of visited node.

$$f(\boldsymbol{\pi}|\boldsymbol{s}) = L(\boldsymbol{\pi}|\boldsymbol{s}) + \lambda(\boldsymbol{\pi}|\boldsymbol{s})$$

$$L(\boldsymbol{\pi}|\boldsymbol{s}) = \sum_{i=1}^{k} ||x_{\pi_i} - x_{\pi_{i-1}}||_2 + ||x_{\pi_1} - x_{n+1}||_2 + ||x_{\pi_k} - x_{n+1}||_2$$

$$\lambda(\boldsymbol{\pi}|\boldsymbol{s}) = \sum_{i \notin \boldsymbol{\pi}} \lambda_i$$

The $L(\boldsymbol{\pi}|\boldsymbol{s})$ is the tour length, and $\lambda(\boldsymbol{\pi}|\boldsymbol{s})$ is the total penalty of the unvisited nodes. The $k = |\boldsymbol{\pi}|$, $k \leq N$ because the entire tour does not contain all of the nodes. There is a constraint of minimum prize $\mu_{(c)}$ as follows:

$$\sum_{i \in \boldsymbol{\pi}} \mu_i \geq \mu_{(c)} \tag{3}$$

Most of MDP is similar with TSP including training scheme. Unlike TSP, there is restriction that the action on depot node $\pi_{N+1}$ is forbidden to be selected until constraint (3) is satisfied. We define cumulative reward for solution (action sequences) from problem instance $s$ as $R = R_f + \alpha R^S$, where $R_f = -f$, $R^S$ is the entropy reward in section 4.1, and $\alpha$ is a hyperparameter.

**Capacitated Vehicle Routing Problem (CVRP).** In CVRP, the vehicle can no longer visit nodes, when it exceeds the maximum demand $v_{(c)}$. Thus, the vehicle has to go back to the depot node, and start another tour. The vehicle can make k number of tours, $\boldsymbol{\pi} = \{\boldsymbol{\pi}_{(i)}\}_{i=1}^{k}$, where the first and last element of each sub-tour (sub-permutation) $\boldsymbol{\pi}_{(i)}$ is the depot node. Let $N$ be the number of nodes. Then the instance of CVRP expressed as $s = \{(x_i, v_i)\}_{i=1}^{N+1}$, where the depot node is $s_{n+1} = (x_{n+1}, 0)$. The objective of CVRP is minimizing $f$:

$$f(\boldsymbol{\pi}|\boldsymbol{s}) = \sum_{i=1}^{k} \sum_{j=1}^{l_i} ||x_{\pi_{(i)_j}} - x_{\pi_{(i)_{j-1}}}||_2$$

The $l_i = |\boldsymbol{\pi}_{(i)}|$. For every tour $\boldsymbol{\pi}_{(i)}$, their is constraint on maximum demand $v_{(c)}$:

$$\sum_{j \in \boldsymbol{\pi}_{(i)}} v_j \leq v_{(c)} \tag{4}$$

The MDP formulation is mostly same as TSP. There is action restriction rule based on constraint (4), where every action except selecting depot node is restricted when it may exceeds the maximum demand (i.e. when it make violation of (4) when selecting node other than depot node).

Similarly to PCTSP, our reward is $R = R_f + \alpha R^S$, where $R_f = -f$. Policy structure and training scheme are the same as TSP.

## A.2 Detailed Implementation of Seeder in Inference Phase

This section provides implementation details of the seeder for the experiments. Our seeder $p^S$ is parameterized by the AM [12], which is the transformer [24] based encoder-decoder model. For the details of the architecture of the AM, see Kool et al. [12].

**SoftMax Temperature.** The output of the AM architecture is the compatibility of the query of all nodes $u_{(c)}$ (see (7) in Kool et al. [12]). Then the probability of selecting nodes can be expressed with the SoftMax function as follows:

$$p_{\theta^S}(\pi_t = i | \boldsymbol{\pi}_{1:t-1}, \boldsymbol{s}) = SoftMax(u_{(c)}, i, T)$$

$$SoftMax(u_{(c)}, i, T) = \frac{e^{\frac{u_{(c)i}}{T}}}{\sum_j e^{\frac{u_{(c)j}}{T}}}$$

The SoftMax temperature $T$ is an important hyperparameter of the sampling of the seeder. Note that if $T \approx 0$ then $p_{\theta^S}(\pi_t = i | \boldsymbol{\pi}_{1:t-1}, \boldsymbol{s})$ will select greedy samples: i.e. $\pi_t = argmax(p_{\theta^S}(\pi_t = i | \boldsymbol{\pi}_{1:t-1}, \boldsymbol{s}))$. If $T \approx \infty$, it will be the same as random search. In the training phase, we set $T = 1$. The details of setting $T$ in the inference phase (i.e. in experiments) is described in Appendix A.5.

## A.3 Detailed Implementation of Reviser

This section describes the detailed implementation of the reviser for each target problem.

**Travelling Salesman Problem (TSP).** Let the reviser($L$) is trained on TSP with $L$ nodes. During the training phase, starting node and end node is restricted to be selected; thus $L - 2$ nodes are in action space.

During the inference phase, the length of segment (the number of nodes in the action space) $l = L - 2$ (the -2 is because starting and end node are restricted), number of segment $K = N/L$ ($N$ : number of nodes of seed from seeder). The starting point of each segment is represented as $\{k, k+L, k+2L, .., \}$, where the every segment is disjoint each other.

The reviser repeats revising process by re-assigning $1 \leq k \leq L$. In the experiments, we simply assigned $k_i = (k_{i-1} + 1)$ iteratively, $1 \leq i \leq I$. See Algorithm 1 for details.

Let $I$ be number of iteration of revision process. For **LCP\*** in Table 1, we used reviser(10). For **LCP\*** in Table 1, and TSPLIB experiment in Appendix G and large-scale TSP ($N = 500$), we used reviser(20) ($I = 25$) and reviser(10) ($I = 20$) see Algorithm 2 for details.

For CVRP and PCTSP, there is depot node, we do not need additional process process to make reviser unlike TSP's depot node itself being the starting node. Therefore we can use seeder (without entropy reward) as reviser. We note that reviser implementation of other problems of VRP variants, TSP variants are straightforward with a slight change in seeder. Also, most of problems can be also revised by TSP reviser as well.

**Prize Collecting Travelling Salesman Problem (PCTSP).** For the small-scale problems ($N = 20, 50, 100$), after the seeder generates seeds (intermediate solutions), the reviser does not select unvisited nodes or drop visited nodes. That is, the selection of visited nodes is fixed and the reviser only tunes the order of the visited nodes. Thus, we use reviser(10) for small scale experiments. The decomposition rule is same as TSP.

For the large-scale problems ($N = 500$), we set $K = 5, L = 100$. We used the seeder trained in PCTSP ($N = 100$) as reviser on large-scale tasks.

**Capacitated Vehicle Routing Problem (CVRP).** For small-scale problems ($N = 20, 50, 100$) we used reviser(10) with $K = 10$. For large-scale problems ($N = 500$) we used seeder trained on CVRP ($N = 100$) as reviser. We set $K = 2$ and $N = 250$.

Solution of CVRP has sub-tours $\boldsymbol{\pi}_{(i)}$; the starting point of each segment is the same as starting point of the sub-tour. For parallelization (make segment length same), we make padding nodes (depot nodes) when the number of nodes in the sub-tour smaller than the assigned segment length $L$, end of sub-tours.

## A.4 Algorithmic Details of LCP.

This section provides a pseudo-code-based explanation of section 4 and Appendix A.3 for clear understanding. The Algorithm 1 is for single reviser, Algorithm 2 is for double reviser (reviser1, reviser2). These algorithms mainly target TSP; application to PCTSP and CVRP is mostly similar (some difference in decomposition method) as described in Appendix A.3.

---

**Algorithm 1** LCP ($M$: sample width, $K$: # of segment, $l$: length of segment, $I$: # of iteration)

---

1: $k = 1, L = l + 2$
2: $\{\boldsymbol{\pi^{(1)}}, ..., \boldsymbol{\pi^{(M)}}\} \sim p^S$
3: **for** $i = 1 : I$ **do**
4:     $B = \{\{\pi_{k:k+L}^{(1)}, ..., \pi_{k:k+KL}^{(1)}\}, ..., \{\pi_{k:k+L}^{(M)}, ...\pi_{k:k+KL}^{(M)}\}\}$ : Decompose to make $B$.
5:     $R = argmax(p^R(B))$ : Make revised segment $R$ by $p^R$.
6:     $\{\boldsymbol{\pi^{(1)}}, ..., \boldsymbol{\pi^{(M)}}\} = $ Composite$(R)$ : Composition to full trajectory solutions.
7:     $k = k + 1$ : Re-assign segment.
8: **end for**
9: $\boldsymbol{\pi^{(*)}} = $ Best$(\{\boldsymbol{\pi^{(1)}}, ..., \boldsymbol{\pi^{(M)}}\})$: Selecting best solution among candidates.

---

**Algorithm 2** LCP* ($M$: sample width, $K_1, K_2, l_1, l_2, I_1, I_2$)

---

1: $k = 1, L_1 = l_1 + 2$ : Initialization for Reviser 1 (R1)
2: $\{\boldsymbol{\pi^{(1)}}, ..., \boldsymbol{\pi^{(M)}}\} \sim p^S$
3: **for** $i = 1 : I_1$ **do**
4:     $B = \{\{\pi_{k:k+L_1}^{(1)}, ..., \pi_{k:k+K_1 L_1}^{(1)}\}, ..., \{\pi_{k:k+L_1}^{(M)}, ...\pi_{k:k+K_1 L_1}^{(M)}\}\}$ : Decompose to make $B$.
5:     $R = argmax(p^{R1}(B))$ : Make revised segment $R$ by $p^{R1}$.
6:     $\{\boldsymbol{\pi^{(1)}}, ..., \boldsymbol{\pi^{(M)}}\} = $ Composite$(R)$ : Composition to full trajectory solutions.
7:     $k = k + 1$ : Re-assign segment.
8: **end for**
9:
10: $k = 1, L_2 = l_2 + 2$ : Initialization for Reviser 2 (R2)
11: **for** $i = 1 : I_2$ **do**
12:     $B = \{\{\pi_{k:k+L_2}^{(1)}, ..., \pi_{k:k+K_2 L_2}^{(1)}\}, ..., \{\pi_{k:k+L_2}^{(M)}, ...\pi_{k:k+K_2 L_2}^{(M)}\}\}$ : Decompose to make $B$.
13:     $R = argmax(p^{R2}(B))$ : Make revised segment $R$ by $p^{R2}$.
14:     $\{\boldsymbol{\pi^{(1)}}, ..., \boldsymbol{\pi^{(M)}}\} = $ Composite$(R)$ : Composition to full trajectory solutions.
15:     $k = k + 1$ : Re-assign segment.
16: **end for**
17: $\boldsymbol{\pi^{(*)}} = $ Best$(\{\boldsymbol{\pi^{(1)}}, ..., \boldsymbol{\pi^{(M)}}\})$: Selecting best solution among candidates.

---

## A.5 Details of Experimental Setting

**Dataset.** We follow the method introduced in Kool et al. [12], random generation of the datasets of TSP, PCTSP, and CVRP based on the provided code[3].

**Runtime comparison.** A single GPU (NVIDIA RTX 2080 Ti) and a single CPU (Intel i7-9700K) are used for all the experiments; with few exceptions. The speed of heuristic solvers in Table 1 is from Kool et al. [12] where it was performed using two CPUs ($2 \times$ Xeon E5-2630).

All the run time described in section 5 is the average time per instance. For the run time measurement in Table 1, we directly reproduce the AM and DRL-2opt with our machine. The implementation of DRL-2opt is based on code[4] provided by the Costa et al. [31]. The results of the NLNS is from code[5], and others are from Kool et al. [12].

For measuring speed, it is essential to set proper evaluation batch size $B$, which is the number of instances solving in parallel. However, it is difficult to make an absolutely fair setting of parallelization.

---

[3]https://github.com/wouterkool/attention-learn-to-route
[4]https://github.com/paulorocosta/learning-2opt-drl
[5]https://github.com/ahottung/NLNS

For example, heuristic methods are mainly performed in CPU, but DRL frameworks are performed in GPU. The parallelization ability of GPU is usually higher than CPU. Hence, DRL frameworks will show fast speed in high $B$.

Moreover, a fair comparison between DRL frameworks is also difficult. The DRL-2opt has merit with parallelization, but it has slow serial speed because they need iterative inferences (about 2000 iterations). In contrast, our LCP scheme has a fast serial speed because our seeder needs only one inference and reviser needs few iteration. Our revision process is parallelizable for a single instance because the reviser solves the decomposed seeds simultaneously.

For the experiments, we give a restriction of $B \leq 100$. Because, from a practical point of view, 10000 instances of the same scale rarely occur at once, we should restrict the evaluation batch size reasonably. Furthermore, many real-world problems require the sequential solving of routing problems: i.e., it is proper to measure speed in low $B$.

For the run time measurements in large scale experiment ($N = 500$), we set $B = 1$ for all of the baselines. The experiments are performed in the same CPU (Intel i7-9700K) and GPU (Nvidia RTX2080Ti), except for the DRL-2opt and NLNS. For DRL-2opt and NLNS, we set $B = 10$ (10% of total number of instances). Also, note that we reproduce the Concorde, Gurobi, OR Tools, ILS and LKH3 in large scale experiment ($N = 500$), based on code provided by Kool et al. [12].

**Temperature scaling.** In the experiments in small scale $N = 20, 50, 100$, we set the SoftMax temperature $T$ of the AM as 1, which is the default setting reported in Kool et al. [12]. For large scale experiment $N = 500$, we tuned $T$ for AM as 0.1. See Table 4 for the setting of $T$ for the LCP scheme.

Table 3: Number of instances in each experiment.

|  | Table 1 | Figure 4 | Table 2 | Table 5 | Figure 5 | Table 6 | Figure 6 |
|---|---|---|---|---|---|---|---|
| Number of Instances | 10000 | 100 | 1000 | 1000 | 100 | 1000 | 10000 |

Table 4: SoftMax temperature of LCP scheme in Table 1, Table 2, and Figure 4.

|  | $n = 20$ | $n = 50$ | $n = 100$ | $n = 500$ |
|---|---|---|---|---|
| TSP | 2 | 2 | 2 | 0.1 |
| PCTSP | 3 | 2 | 1.5 | 0.2 |
| CVRP | 3 | 2 | 1 | 0.3 |

# B   Ablation Study of Scaled Entropy Regularization

This section provides a detailed ablation study of $\alpha$, a hyperparameter of scaled entropy reward $R^S$. We target TSP ($N = 100$) with randomly generated 100 instances. We use reviser(10) with $I = 10$ in the experiment. The SoftMax temperature $T$ of the seeder is fixed as 2. As shown in Table 5, when $\alpha = 0.2, 0.3$, the performance of LCP with uniform scaled entropy regularization exceeds that of linear scaling. However, when $\alpha = 0.4, 0.5$, LCP with linearly scaled entropy regularization outperforms uniform scaling, where they achieve best performances among all cases $\alpha = 0.2, 0.3, 0.4, 0.5$.

Table 5: Cost (lower is better) of LCP scheme with different training hyperparameter $\alpha$ and types of the scaled entropy regularization. The best performances are indicated in bold among the same $\alpha$. The $*$ indicates best performances. The Seeding indicates cost after the seeding process. The Revision indicates cost after the revision process, which is finalized value.

| $\alpha$ | Uniform Scaling | | Linear Scaling | |
|---|---|---|---|---|
| | Seeding | Revision | Seeding | Revision |
| 0.2 | 7.90 | **7.85** | 7.91 | 7.87 |
| 0.3 | 7.91 | **7.86** | 7.91 | 7.86 |
| 0.4 | 7.92 | 7.88 | 7.89 | **7.83*** |
| 0.5 | 7.91 | 7.86 | 7.89 | **7.84** |

# C  Ablation to SoftMax Temperature

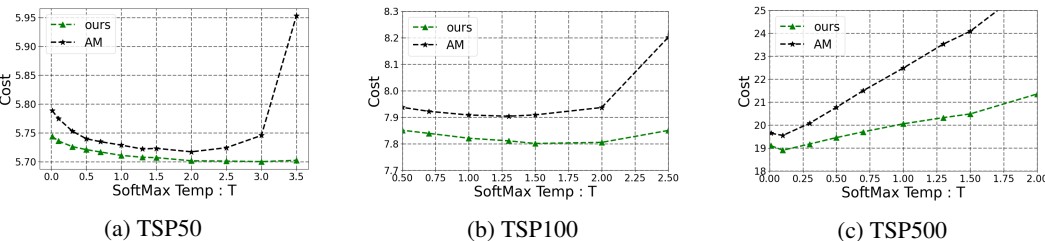

| (a) TSP50 | (b) TSP100 | (c) TSP500 |

Figure 5: The temperature-cost graph of the AM and our scheme on TSP ($N = 50, 100, 500$).

As shown in Figure 5, the experimental results demonstrate that our method is robust to variations on temperature scaling, whereas the AM [12] is vulnerable to high temperature. We remark that the optimal SoftMax temperature of our approach is higher than the AM. For example, in TSP ($N = 50$), the AM performs the best at $T = 2.0$, while our method gives the best performance at $2.0 < T < 3.5$. It demonstrates that while high temperature provides diverse solutions, the quality of the solution is insufficient because the fitness landscape of the AM is steeper than our scheme. In TSP ($N = 500$), both methods have a steep fitness landscape; thus, very low temperature $T = 0.1$ gives a reasonable solution. While our approach gives merit to temperature robustness on large scales with significant performance gain, a policy that has a wider fitness landscape is still needed.

# D    Applying the LCP to Other DRL Frameworks

Section 5 mentions that our LCP scheme can easily be applied to other on-policy DRL frameworks. This section presents empirical validation of LCP's flexibility by using the LCP scheme to pointer network [21, 10] in TSP ($N = 20$). In the experiments, the seeder is parameterized by the pointer network, while the setting of the reviser is similar to previous experiments; we use reviser(10) with five iterations ($I = 5$). The pointer network's implementation is based on code provided by Kool et al. [12], the training hyperparameter is the same as them except we set the batch size as 1024, and $\alpha = 0.5$. The SoftMax temperature $T$ of the seeders is fixed as 1. The dataset of 1000 instances is randomly generated using the same method as previous experiments. As shown in Table 6, the sampling method (Pointer Network {1280}) with our LCP achieves the best performance among all settings, where it gives a significant performance increment compared to the vanilla pointer network.

Remarkably, the sampling method of the seeder itself reduces performance. The cost of the pointer network's sampling method is 7.33, while the greedy method gives 3.95. However, with LCP, the cost of the sampling method drastically reduced, finally exceeding the greedy method even if LCP supports the greedy method. It demonstrates that the main idea of LCP of revising diverse seeds is promising to pointer network. Even if the sampled 1280 seeds are not reasonable solutions, seeds with diversity are revised in parallel. Eventually, the best solution among the revised seeds has outstanding performance.

Table 6: Ablation study of LCP components on TSP ($N = 20$). The optimal gap is measured by comparing it with state-of-the-art solvers. As in Table 2, the Entropy Regularization indicates training the seeder with $R^S$, while the default is the uniform scaling. The Linear Scaling means entropy regularization method with linear weight scheduling. The best performances are marked in bold. The Pointer Network (greedy) indicates Pointer Network with greedy selection, and the Pointer Network {1280} means the sampling method where the sample width is 1280.

| Component of the LCP | | | Pointer Network (greedy) | | Pointer Network {1280} | |
|---|---|---|---|---|---|---|
| Entropy Regularization | Weight Scheduling | Reviser | cost | gap | cost | gap |
| | | | 3.95 | 2.63% | 7.33 | 90.75% |
| ✓ | | | 3.95 | 2.71% | 7.30 | 89.77% |
| ✓ | ✓ | | 3.95 | 2.62% | 7.32 | 90.29% |
| | | ✓ | 3.89 | 1.27% | 3.85 | 0.21% |
| ✓ | | ✓ | 3.89 | 1.18% | 3.85 | 0.24% |
| ✓ | ✓ | ✓ | 3.89 | 1.24% | **3.85** | **0.20%** |

# E   Comparison with Reviser and Improvement Heuristics

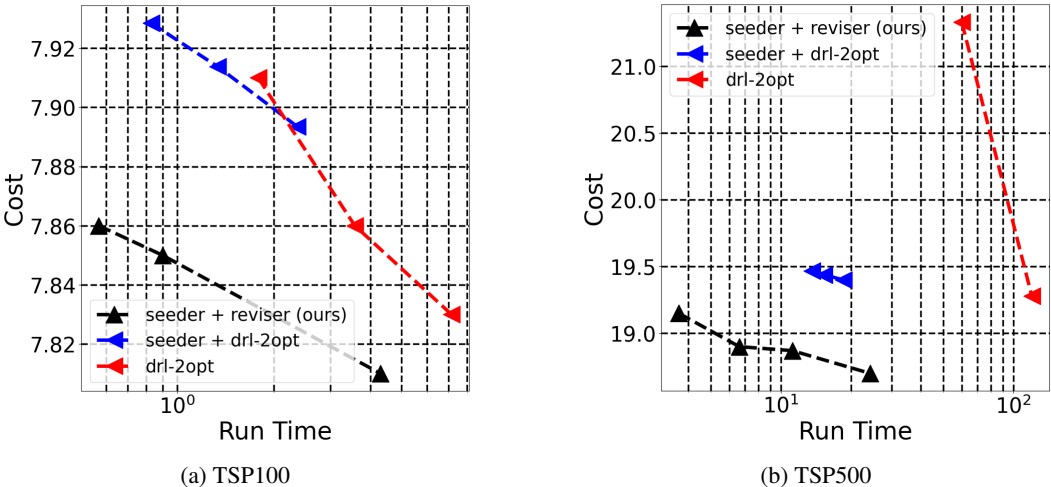

(a) TSP100              (b) TSP500

Figure 6: The time-cost graph on TSP ($N = 100, 500$).

This experiment was conducted to test the reviser's performance. Seeder+reviser outperforms Seeder+DRL-2opt in both time and performance. This experimental result demonstrates that the reviser performs well in a fast and accurate improvement heuristic role. Since reviser can be designed easily by modifying the seeder appropriately, when a new type of seeder (constructive heuristic) is proposed, we can create a high-performance improvement heuristic (i.e., reviser) accordingly.

# F Experiments of Training Convergence in Different PyTorch Seeds

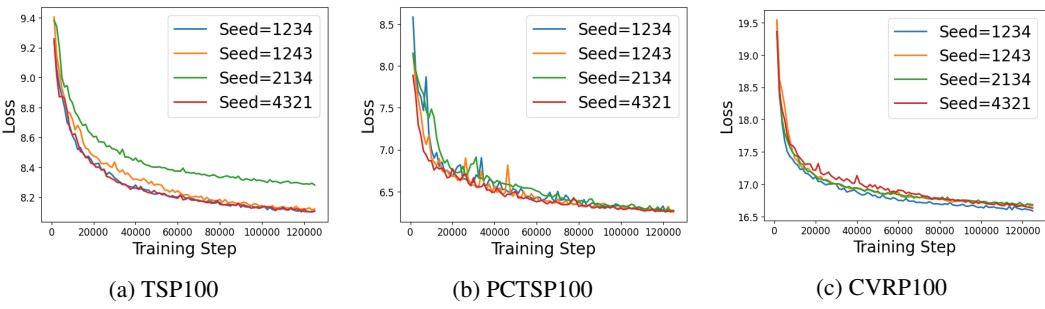

(a) TSP100      (b) PCTSP100      (c) CVRP100

Figure 7: Training graph of the seeder on TSP,PCTSP and CVRP ($N = 100$).

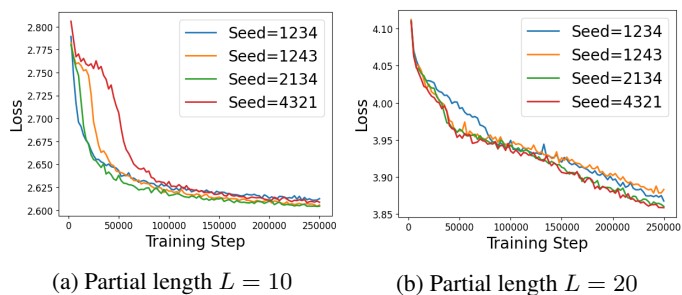

(a) Partial length $L = 10$      (b) Partial length $L = 20$

Figure 8: Training graph of the reviser.

This experiment verifies that the proposed seeder and reviser converge in various PyTorch seeds. DRL algorithms are sometimes unstable in convergence to the random seed; therefore, it is crucial to carry out experiments with random seeds. Through this experiment, we can see that both seeder and reviser converge steadily in 4 random seeds.

# G   Details of Real World Experiments on TSP

This experiment evaluates the solver's performance on 35 instances extracted from TSPLIB. This experiment considers the solver's performance in real-world problems and tests how well the model trained on a fixed scale ($N = 100$) fits in instances with various scales.

In this experiment, we experimented with reviser(20) ($I = 25$), reviser(10) ($I = 20$), and sample width $M = 2560$.

We set the sample width of AM as $M = 40000$ and the number of iterations of DRL-2opt as $I = 2000$.

We outperformed the baseline solver in 22 cases out of 35, and the average optimal gap outperformed Drl-2opt by 2.5%. It also showed an overwhelmingly faster speed compared to the DRL-2opt and AM.

Table 7: Performance comparison in real-world instances in TSPLIB. The (time) indicates time spending. We select the best performance among SoftMax temp $T \in \{0.1, 0.5, 1, 2\}$ in both AM and ours. Therefore, we multiply $4 \times$ to spent time.

| Instance | Opt. | AM [12] | | | DRL-2opt [19] | | | AM + LCP *(ours)* | | |
|---|---|---|---|---|---|---|---|---|---|---|
| | | Cost | Gap | Time | Cost | Gap | Time | Cost | Gap | Time |
| eil51 | 426 | 435 | 2.11% | 13s | **427** | 0.23% | 460s | 429 | 0.73% | 13s |
| berlin52 | 7,542 | 8663 | 14.86% | 14s | 7974 | 5.73% | 460s | **7550** | 0.10% | 13s |
| st70 | 675 | 690 | 2.18% | 23s | 680 | 0.74% | 540s | 680 | 0.74% | 13s |
| eil76 | 538 | 555 | 3.18% | 27s | 552 | 2.60% | 540s | **547** | 1.64% | 18s |
| pr76 | 108,159 | 110,956 | 2.59% | 27s | 111,085 | 2.60% | 540s | **108,633** | 0.44% | 18s |
| rat99 | 1,211 | 1,309 | 8.09% | 44s | 1,388 | 14.62% | 680s | **1,292** | 6.67% | 24s |
| rd100 | 7,910 | 8,137 | 2.87% | 46s | 7,944 | 0.43% | 680s | **7,920** | 0.13% | 26s |
| KroA100 | 21,282 | 23,227 | 9.14% | 46s | 23,751 | 11.60% | 680s | **21,910** | 2.95% | 26s |
| KroB100 | 22,141 | 23,227 | 8.23% | 46s | 23,790 | 7.45% | 680s | **22,476** | 1.51% | 26s |
| KroC100 | 20,749 | 21,868 | 5.40% | 46s | 22,672 | 9.27% | 680s | **21,337** | 2.84% | 26s |
| KroD100 | 21,294 | 22,984 | 7.94% | 46s | 23,334 | 9.58% | 680s | **21,714** | 1.97% | 26s |
| KroE100 | 22,068 | 22,686 | 2.80% | 46s | 23,253 | 5.37% | 680s | **22,488** | 1.90% | 26s |
| eil101 | 629 | 654 | 4.03% | 46s | **635** | 0.95% | 680s | 645 | 2.59% | 26s |
| lin105 | 14,379 | 16,516 | 14.87% | 49s | 16,156 | 12.36% | 680s | **14,934** | 3.86% | 26s |
| pr124 | 59,030 | 63,931 | 8.30% | 68s | **59,516** | 0.82% | 700s | 61,294 | 3.84% | 37s |
| bier127 | 118,282 | 125,256 | 5.90% | 72s | **121,122** | 2.40% | 720s | 128,832 | 8.92% | 37s |
| ch130 | 6,110 | 6,279 | 2.76% | 77s | 6,175 | 1.06% | 790s | **6,145** | 0.57% | 38s |
| pr136 | 96,772 | 101,927 | 5.33% | 84s | 98,453 | 1.74% | 820s | **98,285** | 1.56% | 38s |
| pr144 | 58,537 | 63,778 | 8.95% | 93s | 61,207 | 4.56% | 720s | **60,571** | 3.47% | 43s |
| kroA150 | 26,524 | 28,658 | 8.05% | 102s | 30,078 | 13.40% | 900s | **27,501** | 3.68% | 44s |
| kroB150 | 26,130 | 27,565 | 5.49% | 102s | 28,169 | 7.80% | 900s | **26,962** | 3.18% | 44s |
| pr152 | 73,682 | 79,442 | 7.82% | 101s | **75,301** | 2.20% | 720s | 75,539 | 2.52% | 44s |
| u159 | 42,080 | 50,656 | 20.38% | 111s | **42,716** | 1.51% | 840s | 46,640 | 10.84% | 45s |
| rat195 | 2,323 | **2,518** | 8.14% | 168s | 2,955 | 27.21% | 1080s | 2,574 | 10.81% | 57s |
| kroA200 | 29,368 | 33,313 | 13.43% | 173s | 32,522 | 10.74% | 1,120s | **31,172** | 6.14% | 86s |
| ts225 | 126,643 | 138,000 | 8.97% | 223s | **127,731** | 0.86% | 1,110s | 134,827 | 6.46% | 113s |
| tsp225 | 3,919 | 4,837 | 23.42% | 224s | **4,354** | 11.10% | 1,160s | 4,487 | 14.50% | 113s |
| pr226 | 80,369 | 90,390 | 12.47% | 228s | 91,560 | 13.92% | 940s | **85,262** | 6.09% | 113s |
| gil262 | 2,378 | 2,588 | 8.81% | 306s | **2,490** | 4.71% | 1380s | 2,508 | 5.49% | 134s |
| lin318 | 42,029 | 47,288 | 12.51% | 397s | **46,065** | 9.60% | 1,470s | 46,540 | 10.72% | 158s |
| rd400 | 15,281 | 17,053 | 11.59% | 458s | **16,159** | 8.10% | 1,870s | 16,519 | 8.10% | 209s |
| pr439 | 107,217 | 160,594 | 49.78% | 744s | 143,590 | 33.92% | 1760s | **130,996** | 22.18% | 228s |
| pcb442 | 50,778 | 58,891 | 15.98% | 897s | 57,114 | 12.48% | 1,760s | **57,051** | 12.35% | 228s |
| avg. gap | 0.00% | | 9.90% | | | 7.63% | | | **5.14%** | |