# OpenReview forum: "Learning Collaborative Policies to Solve NP-hard Routing Problems"
_NeurIPS.cc/2021/Conference — NeurIPS 2021 Poster_

### Official Review · Reviewer_AG5Z · 2021-07-14

**Rating:** 6
**Confidence:** 3

**Summary:**

This paper proposes to learn collaborative policies for routing problems. Collaborative policies firstly learn to generate many diversified and primitive solution candidates. Then for each solution candidate, they segment the original problem into several sub-problems. Another neural network model is trained to improve these sub-problems. The improved solutions of the sub-problems can be merged to form a new solution for the original problem.  They teste their methods on the TSP, PCTSP, CVRP problems and achieve the best performance among all the baselines.

**Ethical Concerns:**

N/A.

**Limitations And Societal Impact:**

For the limitations, the authors do not talk about will the method scale to high problems, e.g., TSP-10000. My feeling is that the seeder policy may be not good enough if the size of the sub-problems is increased a lot.

I don't think this work has some bad societal impact.

**Main Review:**

Originality:
I think the method is a smart combination of "Generalize a Small Pre-trained Model to Arbitrarily Large TSP Instances" Fu et al., "Deep Policy Dynamic Programming for Vehicle Routing Problems" Kool et al and "Learning 2-opt Heuristics for the Traveling Salesman Problem via Deep Reinforcement Learning" Costa et al. However, the related work section mentions these two works but does not discuss the relation to these three works.

Quality:

The main contribution of this paper is to make the entire pipeline fully deep learning and do not require any label during the training process. The paper uses a neural network model trained with DRL to generate the primitive solutions and proposes an entropy regularizer term to increase the diversity of the solutions.  Previous works typically supervised train a model to predict the importance of each edge and create the sub-problems (solutions). For the improving phase, this paper uses another neural network model trained through DRL to improve the sub-solutions.  While the previous works leverage DP algorithm, MCTS, or require labels.

The experiment results look good to me. The method beats all the fully neural baselines with reasonable time.

Clarity:

The paper has some important points that I am confused about.

1. How did you generate the sub-problems (segments) of a candidate solution? I assume it is fully random and make sure the points you selected are connected? I think this step is vital in the entire process but it is not explained.
2. For training the revision policy, how did you generate the training instances? It is exactly the same as the one for seeder policy?

Significance:

The method is smart to leverage the previous ideas in fully deep learning and no label required way.

Questions:

1. I think one interesting ablation study can be: what if you use a supervised trained model to generate the segments/sub-problems like what they did in "Generalize a Small Pre-trained Model to Arbitrarily Large TSP Instances". I am wondering how the entropy regularize help compared with a supervised method.

2. For the TSP problems, the size of the sub-problems is 10 vertices. In this case, why a revision model is required since you can compute the optimal solution very fast for only 10 vertices? What're the results then? What if you increase the size of the sub-problems (which can potentially be generalized to huge TSP problems).


**Time Spent Reviewing:**

3.5

---

> ### Author Response · Authors · 2021-08-10
> **Response to Clarity Issues**
>
> Thank you for your comments about clarity.
>
> **Response to Clarity Issues**
>
> **Q1: How did you generate the sub-problems (segments) of a candidate solution? I assume it is fully random and make sure the points you selected are connected? I think this step is vital in the entire process but it is not explained.**
>
> **A1:** The sub-problems are provided randomly; the iteration is done by shifting the cutting points. This process is described by the pseudo-code (see Appendix A.4). The segmenting strategy for producing sub tours is essential, but the randomly-cutting strategy works effectively as the reviser solves the sub-segment of multiple candidate solutions, which are generated with high randomness, and there is also a best-solution-selection mechanism that serves as an improvement step. If we revise the single candidates, the segment rule can possibly affect the entire performance significantly. We will consider a learning-based segmenting strategy in further research.
>
> ***
>
> **Q2: For training the revision policy, how did you generate the training instances? It is exactly the same as the one for seeder policy?**
>
> **A2:** It is exactly the same as the one for seeder policy. We use the randomly generated problem instances whose size is less than 20 (i.e., N=10, 20) to train the reviser.

---

> > ### Author Response · Authors · 2021-08-10
> > **Response to Questions**
> >
> > Thank you for your valuable questions that can improve the paper's quality.
> >
> > **Response to Questions**
> >
> > **Q1: I think one interesting ablation study can be: what if you use a supervised trained model to generate the segments/sub-problems like what they did in "Generalize a Small Pre-trained Model to Arbitrarily Large TSP Instances". I am wondering how the entropy regularizes help compared with a supervised method.**
> >
> > **A1:** We think LCP can be easily applied to the setting of “Generalize a Small Pre-trained Model to Arbitrarily Large TSP Instances” (termed GNN+MCTS [1]). We believe that there are two strategies to employ LCP to GNN+MCTS. First, we can employ the seeding strategy to make the GNN model imitate Concorde’s guiding solution while maximizing the entropy of the policy. After that, the MCTS can perform exploitation; the proposed reviser can also be used to revise the solution provided by MCTS. Second, we can use MCTS as a seeder to generate multiple candidate solutions; we can control the exploration term of the upper confidence bound (UCB) to generate solutions that are as diversified as possible. Then, the AM-based reviser [2] can be employed to improve the candidate solution in the framework of LCP.
> >
> > [1] Fu, Zhang-Hua, Kai-Bin Qiu, and Hongyuan Zha. "Generalize a Small Pre-trained Model to Arbitrarily Large TSP Instances." arXiv preprint arXiv:2012.10658 (2020).
> >
> > [2] Kool, Wouter, Herke Van Hoof, and Max Welling. "Attention, learn to solve routing problems!." arXiv preprint arXiv:1803.08475 (2018).
> >
> >
> > ***
> >
> > **Q2: For the TSP problems, the size of the sub-problems is 10 vertices. In this case, why a revision model is required since you can compute the optimal solution very fast for only 10 vertices? What're the results then? What if you increase the size of the sub-problems (which can potentially be generalized to huge TSP problems).**
> >
> > **A2:** Any reviser can be used for the proposed LCP framework, given that the reviser is fast and efficient in finding a better solution. Thus, when the size of the sub-problem is small, we can use Concorde to compute the optimum solution. However, for the small-sized problem, the AM-based reviser can compute the near-optimum solution with a much faster speed because the reviser can solve multiple problems in a parallel manner. To clearly show how the size of the sub-problems affects the accuracy and computational time of several revisers, we conducted an additional ablation study. The results are summarized below.
> >
> > ***
> >
> > **Table A:** Ablation study of the comparison between the Concorde and our reviser in TSP (N=10) in 10000 randomly generated test problems. The time is spent time for solving 10000 problems.
> >
> > |Method|Cost|Time|
> > |------|---|---|
> > |Concorde|2.51|51s|
> > |Reviser|2.61|1s|
> >
> > ***
> >
> > **Table B:** Ablation study of sub-problems size L in TSP (N=100), I=30 for every L. “Opt Gap” indicates the optimal gap between the optimal solution; the lower is better.
> >
> > |$L$|Opt. Gap.|
> > |------|---|
> > |5 {$I=30$} |1.26%|
> > |10 {$I=30$} |0.85%|
> > |20 {$I=30$} |0.99%|
> > |20 {$I=20$} + 10 {$I=10$} (LCP*)|0.66%|
> >
> > ***
> > As shown in Table A, the reviser achieves a near-optimal solution, having 51 times faster speed than the Concorde. As shown in Table B, solving a larger sub-problem does not guarantee its performance because the reviser in N=20 cannot give a near-optimal solution like the reviser in N=10. If we utilize only a single size segment, N=10 is the best choice for revising. However, when we utilize two revisers, where the first reviser solves N=20 and the second reviser fine-tunes sub-solutions in N=10, the performance is better than the case where only a single reviser solves the segment problem of N=10. That is because the reviser in N=10 can correct local mistakes of the reviser in N=20, making it possible for LCP to search over larger solution space (seeder solves N=100, reviser1 solves N=20, reviser2 solves N=10; termed LCP* in the paper).

---

> > > ### Author Response · Authors · 2021-08-10
> > > **Response to Limitation**
> > >
> > > Thank you for the comment about the possible limitations of the proposed method.
> > >
> > > **Response to Limitation**
> > >
> > > We agree that the AM is not capable of producing near-optimal solutions in large-scale TSP (N=10,000). We think this is because the recurrent decoding scheme of AM degrades significantly when the problem size increases. However, the LCP is not a single method for TSP, but a reusable scheme that can be applied to the existing DRL-based solver. Therefore, the performance of LCP is determined by how it can “accelerate” the existing framework. In that sense, we measured improvement achieved when applying LCP to the vanilla AM in solving large-scale TSP (N=10,000) (See table below). Note that the number of candidate solutions, M, is smaller than other experiments conducted in the paper because of the limitation of hardware resources (VRAM of GPU). The result clearly shows that the LCP scheme improves the performance even when the problem is large (N=10,000).
> > >
> > > ***
> > > **Table C:** Evaluations of LCP in TSP (N=10000). The cost is the average value among ten randomly generated problems.
> > >
> > > ||Cost|Avg. Time|
> > > |------|---|---|
> > > |AM {$M=3$}|141.69|15.32s|
> > > |AM + LCP* {$M=1, I=45$}|119.94|13.40s|
> > > |**Improvement**|**21.75**|**1.92s**|

---

> > > > ### Comment · Reviewer_AG5Z · 2021-08-31
> > > > **Thanks for the rebuttals.**
> > > >
> > > > Thanks for the detailed rebuttals. I have read the authors' responses. Most of my questions and concerns are addressed. So I would like to keep my score unchanged.

---

### Official Review · Reviewer_8gsS · 2021-07-15

**Rating:** 6
**Confidence:** 4

**Summary:**

The paper addressed the hard routing problems based on a hierarchical strategy consists of two DRL policies called the seeder and reviser. Together with the two policies, to assess the diversity of candidate solutions, the entropy-regularized reward is adopted. The revise divides the tour into subtours and optimizes the length of them. Experimental results suggest that the proposed LCP framework works on TSP, PCTSP, and CVRP instances.

**Limitations And Societal Impact:**

Solving various classes in a unified manner (or hybrid manner with existing methods) is a promising work in the literature.

**Main Review:**

As the authors mentioned, the proposed LCP framework is not designed to outperform problem-specific solvers (e.g., Concorde for TSP). Therefore, the authors formulate the routing problems as MDP and the task of DRL-policies is to sequentially design a new task (i.e., select one of the unserved tasks). Since the next task should be taken depends on the current status (and implicitly already assigned tasks), the MDP-based approach seems to be reasonable. In addition, the reward-based study covers various problem tasks.

The idea of the seeder (preparing diverse solutions) and revise (updating solutions) is a kind of a traditional approach to solvecombinatorial problems, but the RL-based formulation seems to work efficiently to various problems. Technically the paper seems to be sound but the structure of the paper is a bit hard to follow although the author revised the paper when submission, because the paper covers branches of classes and showed various experimental comparisons (and most of them are included in Appendix).

### Weights for diverse candidate routes

As far as I understand the proposed formulation correctly, the seeder follows a parameterized policy from the previous work [11]. Therefore, the important part of the LCP seems from the entropy-regularized reward to search diverse areas of all solution spaces, which is supported by the reviser part. Although the entropy is hard to compute exactly, a weighted sum of the selected routings is adopted and the weights on the early stage are set to be higher. This weighting mechanism seems to be interesting to generate diverse solutions.  Although this weighting scheme is a hyper-parameter (or hand-crafted setting), the experimental results indicated that the difference of weights does not affect the result too much (e.g., Table 5 of Appendix B).

My concerns are
- (a) Do fixed parameters in Table 5 (e.g., N of TSP, I=10, T=2, ... ) affect the comparison of the provided results?
- (b) Can we have similar results for real instances? As far as I know, characteristics of random TSP (or other class) instances are hard to tune and in some cases, results are different from those obtained from real instances.
- (c) In Appendix E, the performance of the reviser is evaluated by a kind of ablation test using Drl2-Opt. Did the author evaluate the performance of the seeder? (i.e., some heuristics + reviser) If this kind of comparison is useless for some reason, please clarify it to highlight the advantage of the proposed method (or, for example, Appendix D covers this kind of experiment?).

### Revise

The revision process of candidate seeds is essential for LCP. The key background to design the reviser is solving small routing instances is easy (l = L-2). As noted in Sec.4.2, DRL-2opt searches O(N^2) solution space but the LCP searches O(MK l!).

- (d) I suspect that LCP with smaller l (i.e., L) could generate worse solutions than Drl2-opt. Is this correct or do the authors investigate the case?
- (e) The authors mentioned only Drl-2Opt in Sec.4.2. Since the proposed LCP could be applied to PCTSP and CVRP, some RL-based solvers (e.g., AM, NLNS) should be mentioned on the viewpoint of search space to highlight the advantage of LCP if possible.

In addition to $l$, the parameter $I$ should be carefully examined in my opinion. For PCTSP and CVRP, relatively small $I$ is used (1, 5), while TSP uses $l=10$ an $45$. Why such small values are used? Please add the concept of these used parameter values if possible.

**Time Spent Reviewing:**

4

---

> ### Author Response · Authors · 2021-08-10
> **Response to Concerns (a),(b),(c)**
>
> Thank you for your valuable feedback.
>
> **Answer (a):** If the parameter "I" is large, it gives high performances with lower speed. Thus, the parameter "I" controls the tradeoff between the performance and the computational time. For a fair comparison with other DRL frameworks (especially with AM [1]), we manage "I" small enough so that the total spending time is similar to the baseline.
>
> We have already provided the result showing how the solution-searching time (that is proportional to the parameter “I”) is traded off with the solution quality in Figure 4 and Appendix E (In these figures, the x-axis represents the time for solution finding, which is proportional to I).
>
> According to the reviewer’s comment, we have conducted an additional ablation study to clearly show the impact of parameter “I” on the tradeoff between the optimality gap and the computational time. The results are provided in the table below. The experiment is done with TSP (N=100); 100 problems. The "Opt. Gap." indicates the average optimal gap between the optimal solution and the computed solution; the lower is better. As shown in the table, the opt. gap decreases and the average computation time increases with the parameter "I." For the hyperparameter T, we reported an ablation study (see Appendix C).
>
> |I|Opt. Gap.|Avg. Time.|
> |----|---|---|
> |1|1.88%|0.33s|
> |5|1.24%|0.61s|
> |10|0.91%|1.04s|
> |20|0.86%|1.8s|
> |45|0.81s|3.58s|
>
> ***
> **Answer (b):** We have already evaluated the proposed LCP on TSPLIB benchmark problems whose distribution of the cities is more realistic. Most proposed TSP solvers are evaluated using this benchmark problem set and hence, we did the same thing. The results have been summarized in Appendix G.
> Note that the LCP was trained using random TSP instances and employed to TSPLIB benchmark problems without fine-tuning. Thus, the training hyperparameter was the same for all experimental settings. It is easy to tune the hyperparameter in the inference phase, both on random TSP or real-world TSP. For the SoftMax temperature, T, large T (T=2) is suggested for solving the small-sized TSP (N<100). On the other hand, small T (T<0.3) is suggested for solving large-scale TSP. This empirical rule is validated by the ablation study (see Appendix C). Tuning the number of iterations “I” is more straightforward; if we want to gain more performances, we can simply increase the parameter “I”.
>
> ***
> **Answer (c):** We have already evaluated the performance of the seeder and summarized the results in Table 2 of the main text as a part of the main ablation study. In the experiments, we particularly investigated the effect of each or combination of LCP components on the performance. The fourth row of Table 2 indicates the case of “AM + reviser,” and the last row of Table 2 indicates the case of “seeder + reviser” (the proposed model). To clearly answer the reviewer’s question, we report the partial results of the original table as below. The number represents the optimality gap; thus, lower is better. As shown in the table, when the proposed seeder is used with a reviser, it reduces the optimality gaps for all routing problems.
>
> |Method|TSP ($N=100$)|PCTSP ($N=100$)|CVRP ($N=100$)|
> |----|---|---|---|
> |AM + reviser|1.32%|1.13%|2.86%|
> |seeder + reviser (**ours**)|**0.88%**|**1.02%**|**2.37%**|
>
> In addition, Appendix D investigates whether the proposed seeding concept can be employed in a different network architecture other than AM. It investigates the cases of “PN + reviser” and “PN-based seeder + reviser.”
>
> In short, we have validated the effect of a seeder while considering different baseline architectures in the overall solution finding scheme of LCP.
>
> ***

---

> > ### Author Response · Authors · 2021-08-10
> > **Response to Concerns (d),(e)**
> >
> > **Answer (d):** Reviser takes advantage of searching a solution on restricted solution space provided by the seeder, but the reviser still searches for a solution over larger solution spaces than 2OPT or DRL-2opt. DRL-2opt covers search spaces $O(MN^2)$, while reviser covers $O(MK \times l!)$, where K=N/L and l = L-2. When L=5, we observe that the reviser is not compelling enough. When L=10, the reviser covers a much larger space than DRL-2opt that has higher performances.
> >
> > We have conducted a simple ablation study to investigate the effect of the parameter L (see table below). We will report the results for more extensive ablation studies in the Appendix in the revised manuscript.
> >
> > |Method|Cost|Avg. Time.|Search Space per Single Iteration|
> > |------|---|---|---|
> > |Seeder + DRL-2opt {$I=10$}|7.835|2.28s|$O(MN^2) \approx O(1280 \times 10000)$|
> > |Seeder + reviser {$L=5, I=10$}|7.837|0.85s|$O(MK \times l!) \approx O(1280 \times 20 \times 6)$|
> > |Seeder + reviser {$L=10, I=10$}|7.801|1.66s|$O(MK \times l!) \approx O(1280 \times 10 \times 40320)$|
> >
> > As shown in the table, the Seeder + DRL-2opt has a larger search space than the Seeder + reviser {L=5}. Thus, its performance is slightly better. However, when L=10, the Seeder + reviser covers significantly larger search spaces per iteration, and thus, it outperforms Seeder + DRL-2opt when the same iteration is provided (I=10). Note that the time spending of DRL-2opt is also much larger than the reviser even in the same iteration because running the neural architecture of DRL-2opt is more time-consuming than the reviser’s encoding-decoding processes.
> >
> > ***
> > **Answer (e):** LCP is constructed based upon the AM policy network. We compared the search space between LCP (ours) and 2OPT because the 2OPT is a representative improvement operator that is widely used for solving TSP-like problems. The idea of 2OPT can be extended to the DRL framework to learn the strategy of how agent swaps two cities to improve solutions. This idea is natural; many researchers build RL-based improvement heuristics with 2OPT-like methods [2,3,4]. This idea can also be applied to other routing applications, including the CVRP and the mTSP [2-B]. Therefore, the comparison of the reviser and 2OPT-like methods, including DRL-2opt [2-A], was conducted extensively in this study. Note that paper [2-B] is recently published (2021/07/23) as a journal that extends from the previous conference paper [2-A]. In the journal, this method has been employed to solve TSP variants, including the mTSP and the CVRP (LCP outperforms DRL-2opt in CVRP; we will carefully report the comparison results on different problems in the revised manuscript).
> >
> > The NLNS is a problem-specific improvement heuristic constructed based on DRL. The search space covered by NLNS is $O(MT!)$. The search space of the NLNS is designed to decrease gradually (as the temperature parameter $T$ is cooled down), similar to the simultaneous annealing (SA) concept. Therefore, it is not proper to compare the search space of NLNS and the reviser because the search space of NLNS is not fixed. Moreover, while the combination of the seeder and DRL-2opt is possible, the combination of the seeder and NLNS is not proper because NLNS is designed to explore large space and exploit smaller space by the single model (we separate exploration and exploitation in separate models, seeder and reviser). Therefore, it is natural to compare with a single method of NLNS and seeder-reviser scheme as shown in Table 1 and Figure 4.
> >
> > [1] Kool, Wouter, Herke Van Hoof, and Max Welling. "Attention, learn to solve routing problems!." arXiv preprint arXiv:1803.08475 (2018).
> >
> > [2-A] d O Costa PR, Rhuggenaath J, Zhang Y, Akcay A. Learning 2-opt heuristics for the traveling salesman problem via deep reinforcement learning. In: Asian conference on machine learning. PMLR; 2020. pp. 465–480.
> >
> > [2-B] da Costa, P., Rhuggenaath, J., Zhang, Y. et al. Learning 2-Opt Heuristics for Routing Problems via Deep Reinforcement Learning. SN COMPUT. SCI. 2, 388 (2021). https://doi.org/10.1007/s42979-021-00779-2
> >
> > [3] Wu Y, Song W, Cao Z, Zhang J, Lim A. Learning improvement heuristics for solving the travelling salesman problem. 2019. Available from: arXiv:1912.05784.
> >
> > [4] Fu, Zhang-Hua, Kai-Bin Qiu, and Hongyuan Zha. "Generalize a Small Pre-trained Model to Arbitrarily Large TSP Instances." arXiv preprint arXiv:2012.10658 (2020).

---

> > > ### Author Response · Authors · 2021-08-10
> > > **Answer to additional comment about hyperparameters**
> > >
> > > Thank you for your valuable comment about hyperparameters.
> > >
> > > **Answer to additional comment about hyperparameters**
> > >
> > > When solving PCTSP and CVRP, we use a smaller value of “I” than the value used in solving TSP. This is due to the different characteristics of the solutions for the target problems. Specifically, PCTSP has a starting city and depot city that can be different. On the other hand, TSP has the coinciding starting and destination city; thus, its solution trajectory is represented as a loop. When “I” increases, the seeder of LCP shifts the segmentation point (see pseudocode in Appendix A.4) to construct sub-tours. As the solution of TSP is a loop, we can infinitely shift the segmentation point. In contrast, the solution of PCTSP is not a loop; we cannot shift the segmentation infinitely.
> > >
> > > For the CVRP, there are multiple sub-tours. If we utilize the AM-based TSP solver as a reviser, we must set a small “I” because the size of the sub-tour is typically small (N $\approx$ 10); the reviser is restricted to improve those sub-tours. Suppose we utilize the CVRP solver as a reviser. We can keep increasing “I” because the reviser is not restricted to improve the sub-tour (AM-based CVRP solver can consider CVRP constraints, TSP solver cannot consider constraint). In the paper, we utilize the AM-based TSP solver as a reviser for CVRP (N=100) because the accuracy of the CVRP solver (AM-based reviser) is not promising enough to improve the local solution of CVRP (N=100). However, for CVRP (N=500), we utilize an AM-based CVRP solver as the reviser (see Appendix A.3) because the local solution of CVRP (N=500) is effective enough to improve the solution than it does for a small-sized CVRP. Therefore AM-based CVRP solver significantly helps to improve the local solution.
> > >
> > > Note that the concept of reviser is improving the multiple candidate solutions with smaller iteration compared with conventional improvement operators including 2OPT. As shown in Table 2 (third and last row), the reviser with small iterations reduces the optimality gap significantly. To be clear, we partially report experimental results below (a lower value is better).
> > >
> > > ***
> > >
> > > **Table A:** Experiment results of CVRP ($N=100$)
> > >
> > > |Method|Cost|Opt. Gap.|
> > > |------|---|---|
> > > |Seeder|16.20|2.86%|
> > > |Seeder + Reviser {$I=1$} (**ours**)|16.12|2.37%|
> > >
> > > ***
> > >
> > > **Table B:** Experiment results of PCTSP ($N=100$)
> > >
> > > |Method|Cost|Opt. Gap.|
> > > |------|---|---|
> > > |Seeder|6.07|1.62%|
> > > |Seeder + Reviser {$I=5$} (**ours**)|6.04|1.02%|

---

> > > > ### Comment · Reviewer_8gsS · 2021-08-29
> > > > **Thank you for your comments**
> > > >
> > > > Thank you for your insightful comment replies to my comment.
> > > >
> > > > Some points (e.g., evaluating issues) are much clearly explained by the comments. Particularly, I missed reading and finding some parts of them, although they are explained in the submitted paper. However, with the additional experiments and explanations given by the authors, I'd like to say that the technical contribution of the paper is OK (e.g., I wrote 'Technically the paper seems to be sound but the structure of the paper is a bit hard to follow...' in my comment).
> > > >
> > > > Further, the additional ablation experiments of the seeder and revised seem to be insightful. Particularly, the comparison of the search space and results (i.e., cost) was interesting for me. With respect to PCTSP and CVRP, different problems have different characteristics in general. Now, I can understand the case (i.e., PCTSP uses small $I$ as in Table A) and then conjecture that the concept (the seeder and revised) towards various combinatorial problems can be considered as an interesting approach.
> > > >
> > > > In conclusion, after reading the replies from the author, I'd like to increase my score. Thanks.

---

> > > > > ### Author Response · Authors · 2021-08-29
> > > > > **Thank you for your responses and we would like to clarify the reviewer's comment.**
> > > > >
> > > > > Thank you very much for your response, and we are glad that the reviewer found the response meaningful.
> > > > >
> > > > > I'm really sorry, but we are having a hard time understanding the exact intent of the last sentence of your response.
> > > > >
> > > > > I am wondering if you have additional questions or concerns that we can address or explain. If so, it would be great if you can specify the questions so that we can faithfully answer that question. We are really willing to resolve all of your concerns.
> > > > >
> > > > > If you intended to say that you were satisfied with our responses and would like to increase the score, then we are really delighted. If so, we would like to kindly ask the reviewer to check and edit the review to adjust the score (we noticed that the score has not been changed in the official review).

---

### Official Review · Reviewer_Atn1 · 2021-07-16

**Rating:** 6
**Confidence:** 4

**Summary:**

This paper introduces Learning Collaborative Policies (LCP) for learning to optimize TSP-style routing problems. The goal of this was not to outperform all other optimizers, but to outperform RL optimizers, which are usually less effective than traditional approaches, but also are more scalable to task-variation. This makes this class of optimizers more suitable for real-world problems. LCP uses 2 policies to optimize a TSP: a seeder, and a reviser. The seeder generates many valid but diverse solutions (ensured with an entropy maximization term in the reward). The reviser learns to relax small sections of each seed to create more optimal solutions. Revision is repeated a set number of times and the best solution is given.

**Limitations And Societal Impact:**

Yes. The authors have adequately addressed the limitations and potential negative societal impact of their work

**Main Review:**

Strengths:

•	Elegant idea.
             - Macro-optimization through learned seeding + micro-optimization through revision.
•	Beat baseline RL algorithms in 3 TSP variants with 20, 50, and 100 nodes.
             - Variants seemed representative (educated guess).
•	Real-world applicability + scalability.
             - Some of these applications are mentioned.
•	Figures are well-made and understandable.
•	Ablation study is good.
•	Reproducibility info including hyperparameters is mentioned and indexed in the appendix. Code will also be provided in the final version.

Weaknesses:
•	Unprofessional writing.
            - Most starkly, “policies” is misspelled in the title.
•	At times, information is not given in an easy-to-understand way.
            - E.G. lines 147 - 152, 284 - 289.
•	Captions of figures do not help elucidate what is going on in the figure. This problem is mitigated by the quality of the figures, but it still makes it much harder to understand the pipeline of LCP and its components. More emphasis on that pipeline would help with the understanding.
•	100 nodes seem like a small maximum test size for TSP problems (though this is an educated guess). Many real-world problems have thousands or tens of thousands of nodes.
•	Increase in optimality is either not very significant, or not presented to highlight its significance. It would be better to put the improvement into perspective.
•	Blank spaces in table 1 are unclear.

Opportunities:

•	It would be good to describe why certain choices were made. For example, why is the REINFORCE algorithm used for training versus something like PPO? I presume it has to do with the attention model paper this one iterates on, but clarification would be good.

•	More real-world uses of the algorithm could be included to better understand the societal impact, including details on how LCP could be integrated well.

The paper lacks a high degree of polish and professionalism, but its formatting (e.g. bolded inline subsubsections) and figures are its saving grace. The tables are also well structured, if a bit cluttered --- values are small and bolding is indistinct. This paper does a good job of giving this information and promises open source-code on publication.

Overall, the paper and its presentation have several problems, but the idea seems elegant and useful.


**Time Spent Reviewing:**

3 hours

---

> ### Author Response · Authors · 2021-08-10
> **Response to Weaknesses**
>
> Thank you for your feedback.
>
> **Response to Weaknesses**
>
> + Based on the minor comments, we corrected all typos, edited the captions of the figures to clearly explain the pipeline of the proposed method.
>
> + Responding to the criticism that the proposed method is validated using only small-sized problems (N<100), we would like to kindly remind the reviewer that the method has been tested with the large-scale problem with N=500. The result has been summarized in Figure 4 of the original paper. In addition, we have validated the model with the real benchmark data set of TSPLIB (see Appendix G).
>
> + Regarding the comment that the proposed model's performance improvement is not significant, we would like to highlight that our model actually achieved substantial performance improvement even without changing any neural architecture and learning techniques. To clearly support this argument, we refer to the original results from the main text as follows (Figure 4, PCTSP (N=500)). The table below shows that the proposed LCP solution search scheme reduces both the optimality gap and computational times by more than 20% as compared to AM[1], the basic building block model.
>
>
> |   |Cost|Opt. Gap.|Avg. Time.|
> |------|---|---|---|
> |AM|13.99|24.67%|4.07s|
> |AM + LCP (**ours**)|11.66|3.90%|2.89s|
> |**Improvement**|**2.33**|**21.28%**|**1.18s**|
>
> + We agree that the improvement by the proposed method is not clearly presented. We will report the results for readers to compare the performance easily. The blank in Table 1 indicates the case where there is no need for additional iteration of LCP because the model has already clearly outperformed the baseline DRL models with a significantly faster speed.

---

> > ### Author Response · Authors · 2021-08-10
> > **Response to Opportunities**
> >
> > Thank you for your comment about opportunities for improvement.
> >
> > **O1: Describe why certain choices were made including “Why REINFORCE algorithms are used?”**
> >
> > **A1:** We used REINFORCE algorithm for two reasons.
> >
> > First, we wanted to develop an effective and general solution searching scheme that can work cooperatively with the existing DRL-based solvers. The AM [1] is the primary building block that LCP is constructed upon, but LCP can be used for any other existing DRL-based solvers as well. AM has used REINFORCE as a basic learning method in Kool et al. [1], and thus, we also used REINFORCE as the primary learning method. Please note that, although we could further modify and tune the learning strategy to increase the quality of the solution, we did not do fine-tuning to focus on our research objective. Second, the REINFORCE with rollout baseline and gradient clipping is an empirically effective and adequate learning method compared to PPO.
> >
> > Similarly, most of the choice about deep reinforcement learning and neural networks is from the baseline method [1]. The novelty of our research is to propose a reusable scheme that can accelerate existing DRL solvers by using a simple but effective balancing scheme between exploration and exploitation. In addition, we can improve the performance by tuning the hyperparameters, DRL components, and neural architectures to obtain the “state-of-the-art” in the problem-specific tasks. However, as we mentioned in the paper, our main objective is to develop a general solver that can be constructed upon any DRL-based solver instead of developing a solver that can beat the problem-specific solvers like the Concorde.
> >
> > ***
> >
> > **O2: More real-world uses of the algorithm could be included to better understand the societal impact, including details on how LCP could be integrated well.**
> >
> > **A2:**  This suggestion is indeed valuable. The proposed method can be directly applied to various practical tasks. For example, the proposed method can quickly be adapted to dynamic pickup and delivery problems. By restricting the revisit action with a time window, we can easily train the seeder policy to visit realized delivery locations in minimum time-spending. Similarly, Kool et al. [1] also restricted the action to deal with price collecting constraints. The reviser can be trained just like the seeder but with smaller nodes; thus, it can focus on exploitation.
> >
> > Routing on electrical devices, which has been employed by several studies using AM [2], also can be an excellent example problem that LCP can be employed in. As we have already verified that LCP can improve the performance of AM, LCP is believed to improve the performances of routers for electrical design automation (EDA) tasks.
> >
> > [1] Kool, Wouter, Herke Van Hoof, and Max Welling. "Attention, learn to solve routing problems!." arXiv preprint arXiv:1803.08475 (2018).
> >
> > [2] Liao, Haiguang, et al. "Attention Routing: track-assignment detailed routing using attention-based reinforcement learning." International Design Engineering Technical Conferences and Computers and Information in Engineering Conference. Vol. 84003. American Society of Mechanical Engineers, 2020.

---

### Official Review · Reviewer_BKnt · 2021-07-19

**Rating:** 6
**Confidence:** 2

**Summary:**

Deep reinforcement learning (DRL) has recently been applied to variety of optimization problems, outperforming traditional approaches. This paper investigates the application of DRL to the context of routing problems such as the classical travelling salesman problem. The methodology proposed involves two DRL agents: the seeder, which is tasked with generating candidates for the solution that provide a good coverage of the solution space, and the reviser, which is tasked with attaining better solutions based on the reduced solution space provided by the seeder. To generate diverse candidates, the seeder utilizes entropy regularization reward. The authors present experiments indicating that their framework, called learning collaborative policies (LCP) improves over previous DRL frameworks for various routing challenges (e.g., TSP and capacitated vehicle routing).

**Limitations And Societal Impact:**

No negative societal impact is expected.

**Main Review:**

Thanks you for submitting to NeurIPS 2021.

The paper is reasonably well written and the results are, overall, clearly explained.

I found the approach presented for addressing routing problems (seeder + reviser) interesting.

I wonder what makes routing problems special? Do you expect the same approach to attain good results in other optimization domains? Or is there something about the structure of TSP and the other routing problems discussed that is of special significance here? I encourage the authors to elaborate on this point.

I think that the paper could also benefit from more discussion of why the approach produces good results. Specifically, the seeder-reviser methodology is somewhat reminiscent of traditional local search via genetic algorithms, simulated annealing, etc., where many random solutions are generated and these are combined in various ways to generate better solutions. I realize that the argument is that the generation of such candidate solutions by the seeder is better because of the entropy regularization reward, and that employing DRL by the reviser somehow helps it generalize better, but I would have appreciated more discussion/investigation of this point.

Post author response:

Thank you for your responses to my concerns. I still view this as a borderline paper but am adjusting my score to 6.

**Time Spent Reviewing:**

1.5

---

> ### Author Response · Authors · 2021-08-10
> **Response**
>
> Thank you for your feedback.
>
> **Q1: What makes routing problems special?**
>
> **A1:** Routing problems are a type of NP-hard combinatorial optimization where a sequential order of input arguments strongly affects the quality of the solutions; a slight change in the order of the solution sequence can dramatically change the outcome (scheduling performance). Due to this complexity, employing the DRL framework to solve routing combinatorial optimization tasks is more challenging than that of the non-routing combinatorial optimization tasks.
>
> This paper focuses on the routing problem because (1) it is more challenging, and (2) the problem-solving scheme for routing problems can be applied to solve valuable real-world tasks, including electrical design automation (EDA) [1]. Note that various DRL frameworks have been employed to solve the non-routing combinatorial optimization tasks, and hence, the state-of-the-art performances have already been shown [2].
>
> [1] Liao, Haiguang, et al. "Attention Routing: track-assignment detailed routing using attention-based reinforcement learning." International Design Engineering Technical Conferences and Computers and Information in Engineering Conference. Vol. 84003. American Society of Mechanical Engineers, 2020.
>
> [2] Ahn, Sungsoo, Younggyo Seo, and Jinwoo Shin. "Learning What to Defer for Maximum Independent Sets." International Conference on Machine Learning. PMLR, 2020.
>
> ***
>
> **Q2: Why the proposed approach produces good results?**
>
> **A2:** The proposed approach can effectively find the near-optimum solution using two iterative DRL policies: the seeder and the reviser. We believe that the effectiveness of the proposed LCP scheme is on optimally balancing the role of exploration by the seeder and the role of exploitation by the reviser. The seeder generates candidate solutions that are as diversified as possible and segments them into sub-tours, while the reviser improves each sub-tour solution that is generated by the seeder. Thus, the seeder is dedicated to exploring the full combinatorial action space, and the reviser is dedicated to improving the quality of the candidate solution, focusing on the reduced solution space. This cooperative and synergic solution searching scheme between the seeder and reviser is the key to the proposed method's success.

---

### Author Response · Authors · 2021-08-10
**Common Response**

To begin with, we thank all reviewers for their valuable comments on our paper. We provide author responses one by one.

---

### Decision · Program_Chairs · 2021-09-27

**Decision:**

Accept (Poster)

**Comment:**

This paper proposes Learning Collaborative Policies (LCP) for learning to optimize  routing problems. LCP is  a hierarchical strategy that consists of two RL policies called the seeder and reviser. The seeder generates many valid but diverse solutions (using an entropy maximization term in the reward). The reviser learns to relax parts of  the  seed to create more optimal solutions. This process is repeated and the best solution is given. The authors apply the approach to different routing problems (TSP, PCTSP, CVRP problems) and achieve the best performance among all the baselines. There was some discussion, the reviewers felt the authors made a great effort to address the concerns raised by them, which led to the increase of the score by one of the reviewers, and overall the consensus was that the paper was ok, above the threshold.